# Continual Test-Time Adaptation in Computer Vision: Methods, Benchmarks, and Future Directions

**Sarthak Kumar Maharana**[1]                    *sarthak.maharana@utdallas.edu*
**Shambhavi Mishra**[2*]                          *shambhavi.mishra.1@etsmtl.net*
**Yunbei Zhang**[3*]                              *yzhang111@tulane.edu*
**Shuaicheng Niu**[4]                             *shuaicheng.niu@ntu.edu.sg*
**Taki Hasan Rafi**[5]                            *takihr@hanyang.ac.kr*
**Jihun Hamm**[3]                                 *jhamm3@tulane.edu*
**Marco Pedersoli**[2]                            *marco.pedersoli@etsmtl.ca*
**Jose Dolz**[2]                                  *jose.dolz@etsmtl.ca*
**Yunhui Guo**[1]                                 *yunhui.guo@utdallas.edu*

[1] *The University of Texas at Dallas, USA*
[2] *LIVIA ETS Montreal, ILLS International Laboratory on Learning Systems (ILLS), McGill - ÉTS - MILA - CNRS - Université Paris-Saclay - CentraleSupélec*
[3] *Tulane University, USA*
[4] *Nanyang Technological University, Singapore*
[5] *Hanyang University, South Korea*

**Reviewed on OpenReview:** *https://openreview.net/forum?id=mM3rO3Xw1V*

## Abstract

Deep neural nets achieve remarkable performance when training and test data share the same distribution, but this assumption frequently breaks in real-world deployment, where data undergoes continual distributional shifts. Continual Test-Time Adaptation (CTTA) addresses this challenge by adapting pretrained models to non-stationary target distributions on-the-fly, without access to source data or labeled targets, while mitigating two critical failure modes: catastrophic forgetting of source knowledge and error accumulation from noisy pseudo-labels over extended time horizons. In this comprehensive survey, we formally define the CTTA problem, analyze the diverse continual domain shift patterns that characterize different evaluation protocols, and propose a hierarchical taxonomy that categorizes existing methods into three families: optimization-based strategies (entropy minimization, pseudo-labeling, parameter restoration), parameter-efficient methods (normalization layer adaptation, adaptive parameter selection), and architecture-based approaches (teacher-student frameworks, adapters, visual prompting, masked modeling). We systematically review representative methods within each category and present comparative benchmarks and experimental results across standard evaluation settings. Finally, we discuss the limitations of current approaches and highlight emerging research directions, including the adaptation of foundation models and black-box systems, thereby providing a roadmap for future research in robust continual test-time adaptation. We encourage visiting our repository at https://github.com/sarthaxxxxx/Awesome-Continual-Test-Time-Adaptation

## 1 Introduction

In recent years, deep neural networks have achieved remarkable performance across a wide range of tasks (Silver et al., 2018; Russakovsky et al., 2015; Jumper et al., 2021), largely due to the assumption that the

---

*Equal contribution

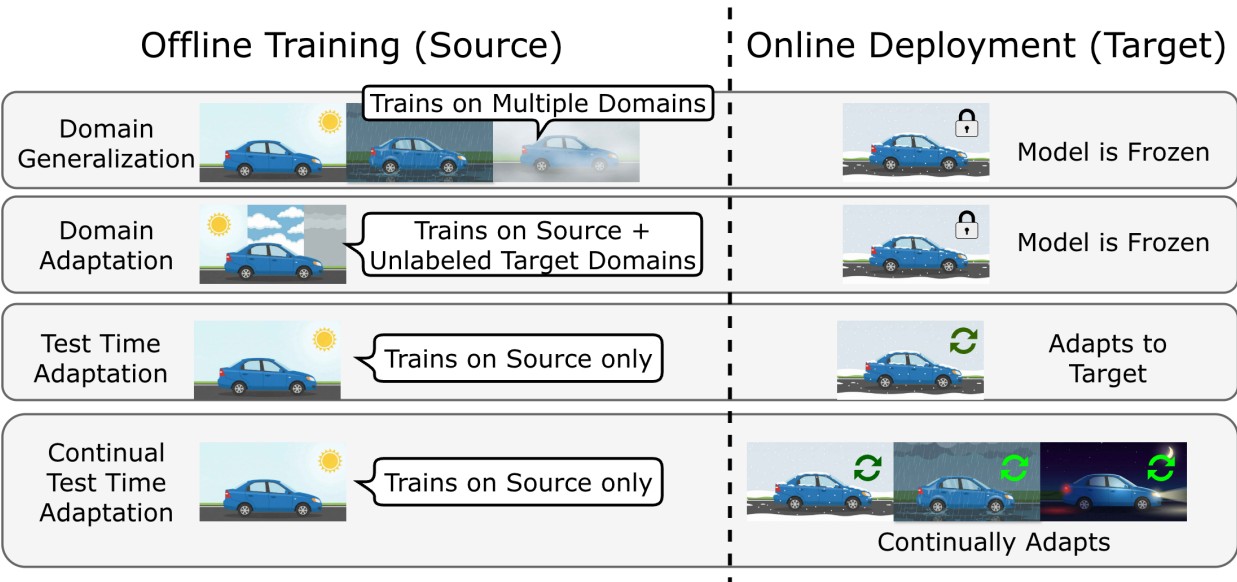

Figure 1: Comparison of adaptation paradigms under distribution shift. **Domain Generalization (DG)** (Zhou et al., 2022) trains on multiple source domains but keeps the model frozen at deployment. **Domain Adaptation (DA)** (Kouw & Loog, 2019) jointly trains on source and unlabeled target data, also resulting in a frozen deployment model. **Test-Time Adaptation (TTA)** (Wang et al., 2021) trains only on source data but adapts to a single target domain during deployment. **Continual Test-Time Adaptation (CTTA)** (Wang et al., 2022) trains on source only and continually adapts to a sequence of evolving target domains at test time, with no access to the domain boundaries. This is indicated by the refresh icons across multiple conditions (snow, rain, night).

training and test data are drawn from the same underlying distribution and that individual samples are independent and identically distributed (i.i.d.) (Goodfellow et al., 2016). This i.i.d. assumption underpins most supervised learning algorithms and is crucial for ensuring that models generalize well from training to unseen/test data. However, in real-world deployment scenarios, this assumption frequently breaks. The data encountered during deployment (test or target data) often differs from the training (source) distribution. These changes, collectively called *distributional shifts*, pose significant challenges for deployed machine learning systems.

Distributional shifts can arise from a variety of sources, ranging from environmental changes to sensor drift to population shifts. For example, consider machine learning models deployed in autonomous or self-driving vehicles (Arnold et al., 2019). During large-scale pretraining, these models may be exposed to clean daylight driving scenarios or similar environments. However, in deployment, they must handle novel conditions such as snow, fog, glare, or those caused by rough weather conditions (Sakaridis et al., 2021). Each of these factors alters the visual statistics of the input data, causing a distributional mismatch. Another example involves visual scenes captured across different camera setups, where shifts in sensor characteristics lead to changes in input distributions (Saenko et al., 2010). Such shifts can degrade the accuracy of the model, lead to unreliable predictions, and significantly hinder generalization to unseen data.

As humans, we naturally adapt and generalize to novel, unseen environments. Inspired by this capability, the machine learning research community has increasingly focused on enhancing model generalization through adaptation strategies to improve robustness against distributional shifts. Traditionally, robustness is addressed during the offline training phase. *Domain Generalization (DG)* (Zhou et al., 2022) leverages multiple source domains to learn invariant representations, aiming for the model to generalize to unseen environments without further modification. Alternatively, *Domain Adaptation (DA)* (Kouw & Loog, 2019) assumes concurrent access to labeled source data and unlabeled target data, jointly optimizing the model

to align distributions. As illustrated in Figure 1, these approaches can be fundamentally categorized by when the adaptation occurs (Offline vs. Online) and the state of the model during deployment (Frozen vs. Adaptive). While effective in specific settings, both DG and DA result in a frozen model at test time. If the target distribution shifts beyond the specific scenarios anticipated during training, the model cannot recover. Furthermore, DA's requirement for simultaneous access to source and target data is often impractical due to data privacy regulations or transmission bandwidth constraints. *Test-Time Adaptation (TTA)* (Wang et al., 2021; Sun et al., 2020) fundamentally alters this workflow by shifting the adaptation process to the online deployment phase. TTA starts with a model pre-trained only on source data. During deployment, it leverages the incoming stream of unlabeled test data to update the model parameters on-the-fly (depicted by the cycle icon in Figure 1). This allows the model to adapt to the specific target domain without ever accessing the source data, preserving privacy and enabling 'source-free' adaptation.

Although standard TTA focuses on adapting a pre-trained source model to a single target domain at test-time, this assumption is often unrealistic in dynamic, real-world settings where models encounter sequences of *non-stationary* and *continually* evolving target domains with rapid shifts in test distributions and no knowledge of task boundaries. In such cases, two critical challenges emerge. The first is *catastrophic forgetting* (Goodfellow et al., 2013): due to continual model parameter updates on long sequences of tasks involving unlabeled data of different distributions, there is a long-term loss of the model's source/pre-trained knowledge. The second challenge is *error accumulation*: as updates occur on noisy test data, errors in early adaptation steps can propagate and compound over time, leading to significant performance degradation. Errors are introduced due to pseudo-labels becoming increasingly noisy and mis-calibrated (Guo et al., 2017).

To address these challenging scenarios, a more general TTA setup involving online continual learning (De Lange et al., 2021; Mai et al., 2022), referred to as ***continual test-time adaptation (CTTA)*** (Wang et al., 2022; Niu et al., 2022; Brahma & Rai, 2023; Maharana et al., 2025a), has gained traction. CTTA extends the standard TTA paradigm by considering non-stationary target distributions that evolve over time, often without explicit task boundaries or access to any labeled data during adaptation. In this setting, models are expected to adapt continuously to a stream of unlabeled test samples, updating

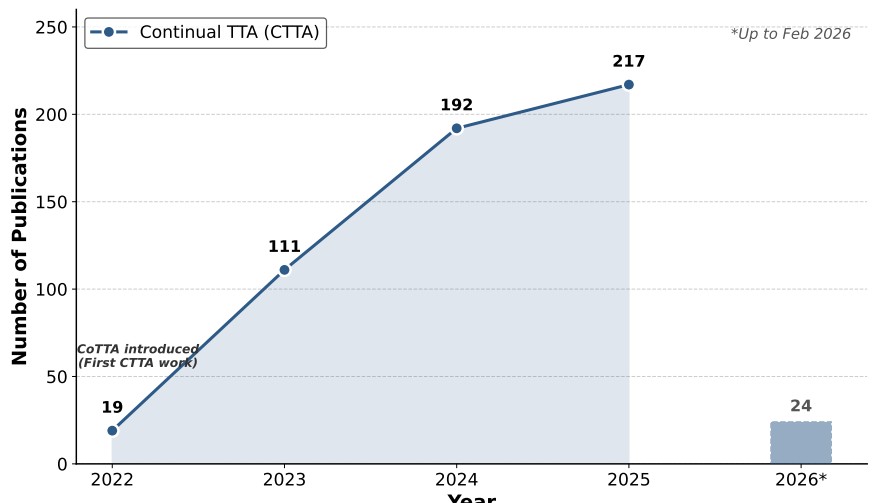

Figure 2: Growth of continual test-time adaptation and related research from 2022 to 2026*. CoTTA (Wang et al., 2022), introduced in 2022, sparked substantial research interest.

their parameters on-the-fly while being robust to distributional shifts. CTTA is performed strictly under test-time constraints, where access to the source data is restricted, the number of updates/iterations per test batch is often limited (strictly online), and computational efficiency is paramount. These constraints make CTTA particularly challenging, as models must remain *plastic* enough to adapt to new distributions, yet *stable* enough to retain knowledge from previously seen domains. CTTA is increasingly relevant in real-world scenarios, such as autonomous driving, medical imaging, and robotic perception, where models are deployed in open-world settings and must learn from streaming data without retraining from scratch. As a result, CTTA serves as a critical step toward building continually adaptive, robust, and deployment-ready models.

As shown in Figure 2, research interest in CTTA has grown substantially since the introduction of CoTTA (Wang et al., 2022) in 2022, reflecting the increasing recognition of its practical importance.

**Need for this survey.** Significant progress has been made in the field recently with respect to both problem formulations and adaptation strategies. However, to the best of our knowledge, there is no comprehensive survey that unifies and systematically discusses the main ideas and contributions of CTTA methods developed over recent years. Such a survey is crucial at this juncture. CTTA sits at the intersection of online continual learning, robustness, generalization, and reliability during model deployment. A careful study of the CTTA literature is essential, particularly as models are increasingly deployed in resource-constrained environments such as edge devices and mobile platforms, where power and memory capacity are limited (Niu et al., 2024). This survey aims to consolidate existing challenges, categorize methods, and identify open research avenues.

**Comparison with related surveys.** Our survey focuses specifically on the continual test-time adaptation (CTTA) setting under distribution shifts. While related surveys (Liang et al., 2025; Xiao & Snoek, 2024; Wang et al., 2024b) on TTA touch upon CTTA, they do so only briefly. Specifically, Liang et al. (2025) provides a comprehensive overview of TTA, including methods, configurations, and a wide range of applications. Similar discussions are found in Xiao & Snoek (2024). In contrast, Wang et al. (2024b) focuses exclusively on online TTA, with CTTA discussed as a subcategory within that context. Our paper differs by focusing specifically and deeply on CTTA, which addresses the more realistic scenario of continually evolving distributions. We believe such an in-depth treatment is pivotal for advancing the field, as it highlights unique challenges and motivates the development of robust methods for dynamic, continually shifting environments.

**Scope.** Throughout this survey, we primarily focus on CTTA methods developed for vision-based models, with an emphasis on recognition. We also discuss extensions to semantic segmentation where relevant. Classification serves as an established testbed with standardized architectures and benchmarks, and the majority of existing CTTA works target this setting. While we acknowledge emerging applications in other domains, including natural language processing (Liu et al., 2025b), medical imaging (Chen et al., 2024), 3D point cloud understanding (Jiang et al., 2024b), and vision-language models (Karmanov et al., 2024), a comprehensive treatment of these areas is beyond the scope of this survey.

**Contributions.** The main contributions of our survey are as follows:

- We provide a formal problem definition of CTTA and systematically analyze the core challenges arising from continual distribution shifts, including catastrophic forgetting and error accumulation.

- We propose a hierarchical taxonomy that categorizes existing CTTA methods into three families: optimization-based, parameter-efficient, and architecture-based approaches, with a detailed discussion of representative methods in each category.

- We summarize standard benchmarks and evaluation protocols and present comparative experimental results across methods.

- We identify limitations of current approaches and highlight emerging research directions, including adaptation of foundation models, vision-language models, and black-box systems.

The rest of this paper is organized as follows. §2 formally defines the CTTA problem setup, introduces notations, and discusses the optimization framework. In §3, we discuss the different settings with respect to the sequence of tasks, especially their dynamic patterns. §4 presents our taxonomy and provides detailed discussion of representative methods in each category. §5 discusses the source model variants often used in the literature. In §6, we talk about the need for online continual adaptation. §7 summarizes benchmarks and experimental comparisons. §8 outlines emerging trends and future research directions. Finally, §9 concludes the survey.

## 2 Preliminaries

In this section, we provide the necessary details of CTTA. This involves the problem setup, notations, and other discussions that would be commonly used throughout this survey. Establishing these foundations ensures a clear and consistent context for analysis.

### 2.1 Problem Definition

CTTA focuses on *continually adapting* a pre-trained source model $f_{\theta^S}$, initially trained on a labeled source distribution $\mathcal{S}$, to a *sequence of unlabeled and evolving target distributions* $\{\mathcal{T}_i\}_{i=1}^K$, where $K$ denotes the total number of underlying (and unknown) shifts or tasks. $\mathcal{T}_i$ denotes the $i^{th}$ task. The task boundaries are not defined. Crucially, CTTA assumes *no access to the source data* during deployment and operates in a strictly *test-time and online* setting (Mai et al., 2022).

At each time-step $t$, the model receives a batch of *unlabeled test samples* from the current distribution, different from the source, and must adapt on-the-fly without revisiting past data or leveraging future information. At time-step $t = 0$, the model parameters are initialized as $\theta^0 = \theta^S$, and updated continually to $\theta^t$ as new batches arrive, with no model reset whatsoever.

*Challenges.* This formulation presents several challenges:

- In this setting, task boundaries are not explicitly defined, so the model remains unaware of when a distributional shift takes place.

- Adaptation must be *efficient*, often limited to a single forward-backward pass per batch.

- Full access to the target domain is not guaranteed; data arrives in *mini-batches*, requiring robust adaptation.

- The absence of ground-truth labels necessitates the use of *unsupervised* or *self-training objectives*, which degrade the performance, especially in extremely noisy conditions.

This setting is similar to models deployed in the real-world, for example, in autonomous/self-driving cars, where models encounter *non-stationary, unlabeled streams/environments* and must *continuously adapt*.

*Goals.* The primary objective of CTTA is to maximize performance on the evolving target data while mitigating two key risks:

- *Catastrophic forgetting* of previously acquired knowledge, including both past target distributions and the source/pre-training knowledge. While standard online continual learning (De Lange et al., 2021) emphasizes preserving performance on earlier tasks during the optimization of parameters on the current task, CTTA introduces an additional challenge: preventing forgetting of the model's source knowledge during continual adaptation. Most methods discussed in this paper largely target the latter.

- *Error accumulation* due to noisy pseudo-labels or unstable parameter updates. Since CTTA operates without access to the ground-truths, many methods are designed to rely on self-generated signals like the pseudo-labels for guidance. However, these signals can be unreliable under severe distribution shifts, which then steer the model parameters further away from the optimal solution. So, a lot of consideration has to be given to carefully handle this.

### 2.2 Continual learning vs. CTTA

Before moving ahead, we find it important to draw clear distinctions between continual learning (CL) (De Lange et al., 2021; Mai et al., 2022) and CTTA. Both learning paradigms involve a model encountering a sequence of tasks and parameter updates without revisiting earlier data. However, there are major distinctions that we discuss below. In Table 1, we provide a summary.

1. Supervision: Standard CL assumes access to ground-truth labels for each task. The model receives labeled pairs and is trained with a supervised loss. In CTTA, there is a strict label absence since the learning is at test-time. The updates rely on self-supervision.

Table 1: Comparison of standard Continual Learning (CL) and Continual Test-Time Adaptation (CTTA).

| Dimension | Standard CL | CTTA |
|---|---|---|
| Supervision | Labeled data pair per task | Unlabeled test data only |
| Loss signal | Supervised (cross-entropy, etc.) | Self-generated (entropy, pseudo-labels, etc.) |
| Source data access | Available during training | Unavailable at test-time |
| Replay/Buffer | Permitted (experience replay) | Not permitted or severely restricted |
| Passes per task | Multiple epochs | Single forward-backward pass |
| Task boundary | Often known | Unknown; must be inferred |
| Forgetting target | Performance on past *training* tasks | Generalization of the *source model* |
| Stability anchor | Previous training task checkpoints | Fixed pre-deployment source model $f_{\theta^S}$ |

2. The role of the source model: In standard CL, a randomly initialized model is trained from scratch or fine-tuned sequentially across incoming tasks, with training on earlier tasks constituting a part of the learning process. However, in CTTA, the source model $f_{\theta^S}$ is fixed before any test data is observed. Clearly, $f_{\theta^S}$ represents the entirety of prior knowledge. Adaptation at test-time is purely corrective and compensates for the distributional shift between the source and target distributions. So, forgetting in CTTA is qualitatively different: what is lost is not performance on a previous *training* task, but the generalization properties that $f_{\theta^S}$ was validated to have before deployment, i.e., the source knowledge.

3. Data access and replay: Standard CL methods commonly address forgetting through experience replay (Lopez-Paz & Ranzato, 2017; Rolnick et al., 2019; Chaudhry et al., 2019). This is unavailable in CTTA. Source data is inaccessible by assumption, motivated by privacy and data constraints. The absence of source data is a defining constraint.

4. Compute regime: Standard CL methods are permitted multiple training epochs per task, with full forward and backward passes and access to the entire task dataset. On the other hand, CTTA operates in a strict online setting where each test batch is observed exactly once, and updates are limited to a single forward-backward pass. A few methods, like FOA (Niu et al., 2024) propose backpropagation-free algorithms.

5. Task boundary information: Most standard CL methods are permitted to know when a task boundary occurs. However, in CTTA, the transition of task boundaries ($\mathcal{T}_i \to \mathcal{T}_{i+1} \cdots$) is unknown, and the model must detect and respond to shifts implicitly.

6. Objective asymmetry: Standard CL balances two objectives: *plasticity* (learning new tasks) and *stability* (retaining performance on older tasks). In CTTA, plasticity is achieved by unsupervised adaptation to the current test/target distribution, while stability is operationalized as preservation of the source model $f_{\theta^S}$.

## 2.3 Notations

Assuming there are a total of $N$ source distributions, we denote the collection as $\mathcal{S} = \{p(x_s^{(k)}, y_s^{(k)})\}_{k=1}^N$. Each distribution $p(x_s^{(k)}, y_s^{(k)})$ corresponds to a particular source domain with its own marginal and conditional characteristics. In the standard CTTA setting, one does not have access to any distribution in $\mathcal{S}$. However, some methods do obtain access to a very small amount of source data to guide adaptation (Niu et al., 2022; Gong et al., 2022; Zhang et al., 2025c;d; Wang et al., 2025c). Now, let, at time-step $t$ of the $i^{th}$ task, the target distribution be denoted as $\mathcal{T}_i = p(x_t, y_t)$. This is defined over the joint space $\mathcal{X} \times \mathcal{Y}$, where $\mathcal{X}$ refers to the feature space and $\mathcal{Y}$ is the label space. Here, for clarity, $(x_s, y_s)$ and $(x_t, y_t)$ refer to a data pair sampled from $\mathcal{S}_k$ and $\mathcal{T}_i$, respectively. A discriminative model $f_{\theta^S} : \mathcal{X} \to \mathcal{Y}$, pre-trained on one or more source distributions, is adapted in the presence of distributional shifts. At test time, the model receives a stream of *unlabeled* test data $\{x_t\}_{t=1}^\infty$, of a single $\mathcal{T}$. The model is adapted online, yielding $f_{\theta^t}$ with parameters $\theta^t$ at each time-step $t$, which evolves as new target inputs are encountered. The model has to be continually adapted to $K$ tasks, with each task having a unique test distribution, *without any reset in model parameters.*

Table 2: A comparison of distributional shifts at test-time, where often $p(x_s, y_s) \neq p(x_t, y_t)$. Let $(x_s, y_s) \sim \mathcal{S}$ and $(x_t, y_t) \sim \mathcal{T}_i$ of the $i^{\text{th}}$ task; shifts are defined by which factors of the joint $p(x, y)$ differ between $\mathcal{S}$ and $\mathcal{T}_i$. In this survey, we focus on covariate shifts only.

| Shift | Condition | Example |
|---|---|---|
| Covariate | $p(x_s) \neq p(x_t), p(y_s\|x_s) = p(y_t\|x_t)$ | Trained on clear-weather driving; deployed in fog/snow. Same classes, altered visual appearance. |
| Concept | $p(x_s) = p(x_t), p(y_s\|x_s) \neq p(y_t\|x_t)$ | Road-surface features predicted "safe" when dry but same features indicate "unsafe" when wet at test-time. |
| Conditional | $p(y_s) = p(y_t), p(x_s\|y_s) \neq p(x_t\|y_t)$ | "Car" class: predominantly sedans in training, but trucks at test-time. |
| Label | $p(y_s) \neq p(y_t), p(x_s\|y_s) = p(x_t\|y_t)$ | Balanced healthy/diseased in training; deployed in a high-prevalence screening region. |

## 2.4 Distributional Shifts

As discussed so far, adapting models at test-time is essential because the source distribution often differs from the target distribution encountered during deployment. Formally speaking, in the CTTA setting, the underlying $i^{th}$ target distribution $\mathcal{T}_i$ may differ significantly from the source, often characterized by a distributional mismatch $p(x_s, y_s) \neq p(x_t, y_t)$, where $(x_s, y_s)$ and $(x_t, y_t)$ are sampled from the source and target distribution respectively. This discrepancy results in poor test generalization of the source model $f_{\theta^S}$ when applied directly to samples from $\mathcal{T}_i$ (Quiñonero-Candela et al., 2008; Koh et al., 2021).

To drive further discussion on distributional shifts, we can decompose the joint distribution $p(x, y)$ based on a Bayesian factorization: $p(x, y) = p(y)\,p(x|y)$ or $p(x)\,p(y|x)$. This decomposition gives rise to several distinct types of distributional shifts commonly discussed (Xiao & Snoek, 2024):

1. We encounter *covariate shift* when $p(y_s|x_s) = p(y_t|x_t)$ but $p(x_s) \neq p(x_t)$, i.e., only the label semantics/spaces are the same.

2. *Concept shift* involves the opposite, i.e., $p(y_s|x_s) \neq p(y_t|x_t)$ but $p(x_s) = p(x_t)$.

3. In *conditional shifts*, $p(y_s) = p(y_t)$ but $p(x_s|y_s) \neq p(x_t|y_t)$. This means that, with the same label space, the difference lies in the input distribution that varies based on the labels.

4. *Label shift* involves shifts in label space, but the label-conditioned distribution remains the same. That is, $p(y_s) \neq p(y_t)$ and $p(x_s|y_s) = p(x_t|y_t)$.

We provide a summary of the mentioned distributional shifts in Table 2 and give certain real-world examples.

**On Semantic Shift and Scope.** A fifth type of shift, semantic shift, is well established in the out-of-distribution (OOD) detection (Hendrycks et al., 2021a) and open-world recognition literature (Bendale & Boult, 2015; Li et al., 2024). Semantic shift refers to classes encountered at test-time that were completely unseen during training. It is categorically distinct from the four shifts above, all of which assume a *closed-set* condition: source and target share the same label space, and the distributional shift concerns only how inputs, labels, or their relationships are distributed within that shared space. In the context of this survey, we emphasize that CTTA literature and the methods surveyed herein predominantly address the *covariate shift* setting. Semantic shift is explicitly outside the scope of this survey. Most CTTA works herein operate under the closed-set assumption. However, we do observe a rise in standard open-set TTA methods (Lee et al., 2023a; Gao et al., 2024; Yu et al., 2024; Dong et al., 2025) and joint covariate and label shifts (Park et al., 2023; Xiao et al., 2024). Extending CTTA to the open-set regime, where semantic shift may co-occur with covariate shift, is an important and largely open research direction. DOCO (Yang et al., 2026) is the first work to introduce this paradigm.

## 2.5 Optimization and Discussions

Assume the model parameters at time-step $t$ are denoted as $\theta^t$ and $\mathcal{L}$ denotes a self-training loss, due to lack of label supervision. Since $\theta^t$ is adapted or adjusted to the test batch $x_t$ of a certain task, the specific model parameters are updated to $\theta^{t+1}$ as:

$$\theta^{t+1} = \theta^t - \alpha \nabla_\theta \mathcal{L}(x_t; \theta)\Big|_{\theta=\theta^t}, \tag{1}$$

where, at $t = 0$, $\theta^t = \theta^S$. Here, $\alpha$ is the learning rate and $\nabla_{\theta^t}\mathcal{L}(x_t; \theta^t)$ denotes the gradient computed based on current test batch $x_t$ and parameters $\theta^t$. This learning rule happens *continually*, unless mentioned otherwise, without any model reset to its source parameters $\theta^S$.

*Key Aspects.* The above update rule embodies several key aspects:

1. *Iterative continual adaptation*: The model is updated sequentially with each arriving test batch. This iterative process allows for immediate adaptation to new data distributions but also poses challenges related to stability over a long sequence of tasks due to noisy gradients.

2. *Loss design*: In CTTA, the loss $\mathcal{L}$ is typically crafted to exploit unsupervised signals from the test data. Its effectiveness determines the model's ability to generalize despite rapid distribution shifts.

3. *Source-free constraint*: Since the source data is unavailable during adaptation, the model must continuously learn from new test samples without overfitting to any particular distribution.

Overall, CTTA critically hinges on the careful design of $\mathcal{L}$ that manages the trade-off between rapid adaptation and long-term generalization in a dynamic, streaming environment. Approaches emphasize the design of $\mathcal{L}$, as the choice of loss function profoundly influences the adaptation dynamics.

## 3 Continual Domain Shift Patterns

In this section, we examine a fundamental aspect of CTTA: the temporal structure in which domains or tasks arrive during deployment. Assumptions about domain ordering, duration, transition dynamics, and sample-level distributional properties give rise to distinct evaluation settings, each presenting unique challenges for adaptation methods. We organize the existing settings from the simplest structured scenarios to increasingly dynamic and realistic ones.

**Continual Structured Change (CSC) is the standard and most widely adopted evaluation setting in CTTA.** Introduced by CoTTA (Wang et al., 2022), CSC assumes that target domains arrive in a fixed sequential order, where each domain $\mathcal{T}_i$ persists for a uniform number of $n$ samples before transitioning abruptly to the next domain $\mathcal{T}_{i+1}$. Formally, given a sequence of $K$ tasks, the test stream $\{x_t\}_{t=1}^{K \cdot n}$ is structured such that all samples $x_t$ for $t \in [(i-1)n+1, \ in]$ are drawn independently from $\mathcal{T}_i$. In common classification benchmarks such as CIFAR-10-C, CIFAR-100-C, and ImageNet-C (Hendrycks & Dietterich, 2019), each of the $K = 15$ corruption domains spans $n = 10,000$, $10,000$, and $5,000$ test samples, respectively. Many subsequent works (Liu et al., 2024b; Park et al., 2024a; Maharana et al., 2025a; Park et al., 2024b; Liu et al., 2024a) adopt this protocol, often extending it to a *multi-round* variant where the full sequence of $K$ domains is repeated for $R$ rounds (e.g., $R = 10$), yielding a total test stream of $R \cdot K \cdot n$ samples. While the multi-round protocol accumulates substantially more domain transitions and tests long-term stability (Duan et al., 2025), it retains the structural properties of CSC with uniform domain durations and predictable ordering (Zhang et al., 2025d). CSC enables controlled and reproducible evaluation; however, its clean boundaries and fixed durations do not fully capture the complexity of real-world deployment, where domain changes are typically irregular and unpredictable.

**Gradual domain transitions relax the assumption of abrupt, instantaneous boundaries between consecutive domains.** RDumb (Press et al., 2023) introduced the Continuously Changing Corruptions (CCC) benchmark, where corruption types evolve smoothly over a very long timescale. Rather than switching instantaneously from $\mathcal{T}_i$ to $\mathcal{T}_{i+1}$, the corruption intensity interpolates gradually between adjacent domains,

producing a continuous stream without sharp boundaries. Similarly, BECoTTA (Lee et al., 2024a) proposed the Continual Gradual Shifts (CGS) benchmark, where the domain identity transitions based on domain-dependent continuous sampling probabilities: the probability of sampling from $\mathcal{T}_i$ decreases over time while the probability of sampling from $\mathcal{T}_{i+1}$ increases, yielding blurred boundaries and mixed-domain batches near transitions. Both CCC and CGS add realism by removing the artificial abruptness of CSC. However, as noted by Zhang et al. (2025d), these settings still follow the overall CSC pattern in that each domain persists for a long duration and the ordering of domains remains predictable. The mixed-domain batches near transitions constitute only a small fraction of the total data.

**Beyond domain-level structure, several works address sample-level distributional challenges that arise in practical deployment.** SAR (Niu et al., 2023) studied TTA under what are termed 'wild' test conditions, identifying three practical challenges: (1) *mixed distributional shifts*, where a batch $B_t$ at time-step $t$ may contain samples from multiple domains simultaneously, i.e., $x \in B_t$ may be drawn from different $\mathcal{T}_i$; (2) *small batch sizes*, including the extreme case of single-sample adaptation where $|B_t| = 1$; and (3) *online imbalanced label distribution shifts*, where the ground-truth test label distribution $Q_t(y)$ is non-stationary and may become imbalanced at each time-step $t$. While SAR was originally proposed for single-domain TTA, these challenges naturally extend to CTTA and significantly affect methods that rely on batch-level statistics for normalization and adaptation. Separately, NOTE (Gong et al., 2022) identified that real-world test streams are often *temporally correlated* (non-i.i.d.). Instead of the standard i.i.d. assumption where $(x_t, y_t) \sim P_T(x, y)$ from a time-invariant target distribution $P_T$, NOTE considers a time-dependent formulation $(x_t, y_t) \sim P_T(x, y \mid t)$, where the marginal label distribution $p(y \mid t)$ varies over time due to temporal correlation. For example, in autonomous driving, consecutive frames are dominated by the same object categories. This non-i.i.d. nature biases batch normalization statistics and degrades entropy-based adaptation objectives.

**Practical Test-Time Adaptation (PTTA) unifies the challenges of continually changing distributions and temporally correlated sampling into a single evaluation protocol.** Proposed by RoTTA (Yuan et al., 2023), PTTA captures compound real-world difficulties where the test distribution changes continually across time-steps while the received batch $\mathcal{X}_t$ at each step contains highly correlated samples rather than i.i.d. draws. To systematically control the degree of temporal correlation, RoTTA employs a Dirichlet distribution parameterized by $\delta$ to govern the *within-batch label composition*. Specifically, the label proportions within each batch are sampled from $\mathrm{Dir}(\delta \cdot \mathbf{1}_C)$, where $C$ is the number of classes. Smaller values of $\delta$ produce batches dominated by a single class (high temporal correlation), while larger values yield more balanced, near-i.i.d. batches. The combination of within-batch label correlation and continually shifting domains creates a particularly difficult setting: batch normalization statistics become unreliable due to biased samples, and adaptation losses such as entropy minimization risk overfitting to the skewed label distribution of the current batch.

**Continual Dynamic Change (CDC) addresses the limitations of uniform domain durations by introducing varying domain frequencies and durations.** Proposed by DPCore (Zhang et al., 2025d), CDC relaxes the core CSC assumption that each domain persists for a fixed number of samples. In CDC, domains may recur multiple times with different durations, and domain transitions can occur more frequently and unpredictably. To control the degree of dynamism, Zhang et al. (2025d) employ a Dirichlet distribution with parameter $\delta$ to govern domain transitions at the *task level*, in contrast to RoTTA's *sample-level* usage. The domain assignment probabilities are sampled from $\mathrm{Dir}(\delta \cdot \mathbf{1}_K)$, where $K$ is the number of domains. Smaller values of $\delta$ produce longer, more stable domain segments resembling CSC, while larger values lead to rapid and frequent domain switches. For instance, at $\delta = 1$, the setting maintains a moderate balance between structure and randomness, while at $\delta = 10$, domain transitions become extremely frequent, creating a highly dynamic environment. CDC exposes three critical limitations of existing CTTA methods that are less apparent under CSC: (1) *convergence difficulty* due to insufficient samples within brief domain exposures, (2) *catastrophic forgetting* as model parameters are continually overwritten across many rapid domain changes, and (3) *negative transfer* when knowledge acquired from one domain adversely affects adaptation to dissimilar domains. Moreover, Zhang et al. (2025d) demonstrated that CDC naturally compounds with sample-level challenges: when domain durations are very short, batches near transitions contain samples from multiple domains (mixed distributional shifts), and the rapid domain changes induce label imbalance within individual

batches. Evaluating under the combination of CDC with the temporal correlation of PTTA further amplifies these difficulties, as the model must simultaneously handle unpredictable domain dynamics and biased within-batch label distributions.

**Recurring and evolving domain patterns capture the long-term, non-stationary nature of real-world deployment where previously encountered conditions reappear.** ReservoirTTA (Vray et al., 2025) formalized three complementary domain shift patterns for prolonged test-time adaptation. *Recurring Continual Structured Change* follows a predictable domain order where domains periodically reappear, i.e., the sequence $\mathcal{T}_1 \rightarrow \mathcal{T}_2 \rightarrow \cdots \rightarrow \mathcal{T}_K$ is repeated cyclically. Unlike the multi-round CSC protocol of Wang et al. (2022), the recurring formulation emphasizes the model's ability to *detect* returning domains and *reuse* previously learned knowledge rather than re-adapting from scratch. *Recurring Continual Dynamic Change* involves unpredictable domain shifts where certain domains recur but without a fixed order, leading to irregular and abrupt transitions with no guarantee on when or for how long a domain will reappear. *Continuously Changing Corruptions* describes scenarios where conditions within a domain evolve incrementally (e.g., weather gradually intensifying) before eventually transitioning to an entirely new domain. These patterns are particularly relevant for long-horizon deployments such as autonomous driving and environmental monitoring, where visual conditions shift and recur over extended periods. The explicit treatment of domain recurrence introduces an important dimension: the model must not only adapt to novel domains but also efficiently re-adapt to previously encountered ones, ideally leveraging domain knowledge retained from prior exposures rather than re-learning from scratch.

Overall, these diverse formulations reveal a rich spectrum of evaluation settings in CTTA, ranging from the controlled and predictable CSC to the dynamic CDC and recurring domain patterns. Importantly, these settings are not mutually exclusive. In real-world deployment, multiple challenges are likely to co-occur: domains may change dynamically with varying durations (CDC) while test samples within each domain are temporally correlated (PTTA), batches may contain mixed-domain data (wild TTA), and previously seen conditions may recur unpredictably (Recurring CDC). The compounding of these factors creates a considerably more demanding adaptation problem than any individual setting alone. As domain changes become more frequent, less predictable, or more gradual, CTTA methods face increasing pressure to balance rapid plasticity with long-term stability. Evaluating CTTA methods under this full range of domain shift patterns, and especially their combinations, is therefore essential for developing adaptation strategies that remain robust and reliable in realistic, non-stationary deployment environments.

## 4 Taxonomy of Continual Test-Time Adaptation Methods

We divide the discussion on CTTA methods based on what they adapt and their update strategies. Figure 3 provides a hierarchical overview of the taxonomy.

1. **Optimization-based**: As discussed in §2.5, the choice of a self-training loss objective $\mathcal{L}$ has a strong impact on the objective. Model predictions can be unreliable without supervision and under distributional shifts, leading to noisy gradients. To address this, we discuss three key paradigms: a) entropy minimization, which encourages confident predictions; b) pseudo-labeling, which leverages high-confidence predictions as surrogate supervision; and c) parameter restoration, which helps in mitigating forgetting by recovering source knowledge.

2. **Parameter-Efficient**: CTTA methods primarily approach adaptation from two complementary perspectives. One line of work focuses on estimating the normalization statistics of BatchNorm (Ioffe & Szegedy, 2015) layers *w.r.t.* the test distribution. Another group of work emphasizes adaptively selecting suitable layers to adapt, often inspired by findings in transfer learning literature (Weiss et al., 2016).

3. **Architecture-based**: A parallel line of CTTA research employs teacher-student frameworks to enhance adaptation stability. The teacher model typically produces stable pseudo-labels to guide the student to adapt. In addition, a few CTTA works also introduce domain adapters, visual prompting (Bar et al., 2022; Jia et al., 2022), and masked image modeling (He et al., 2022).

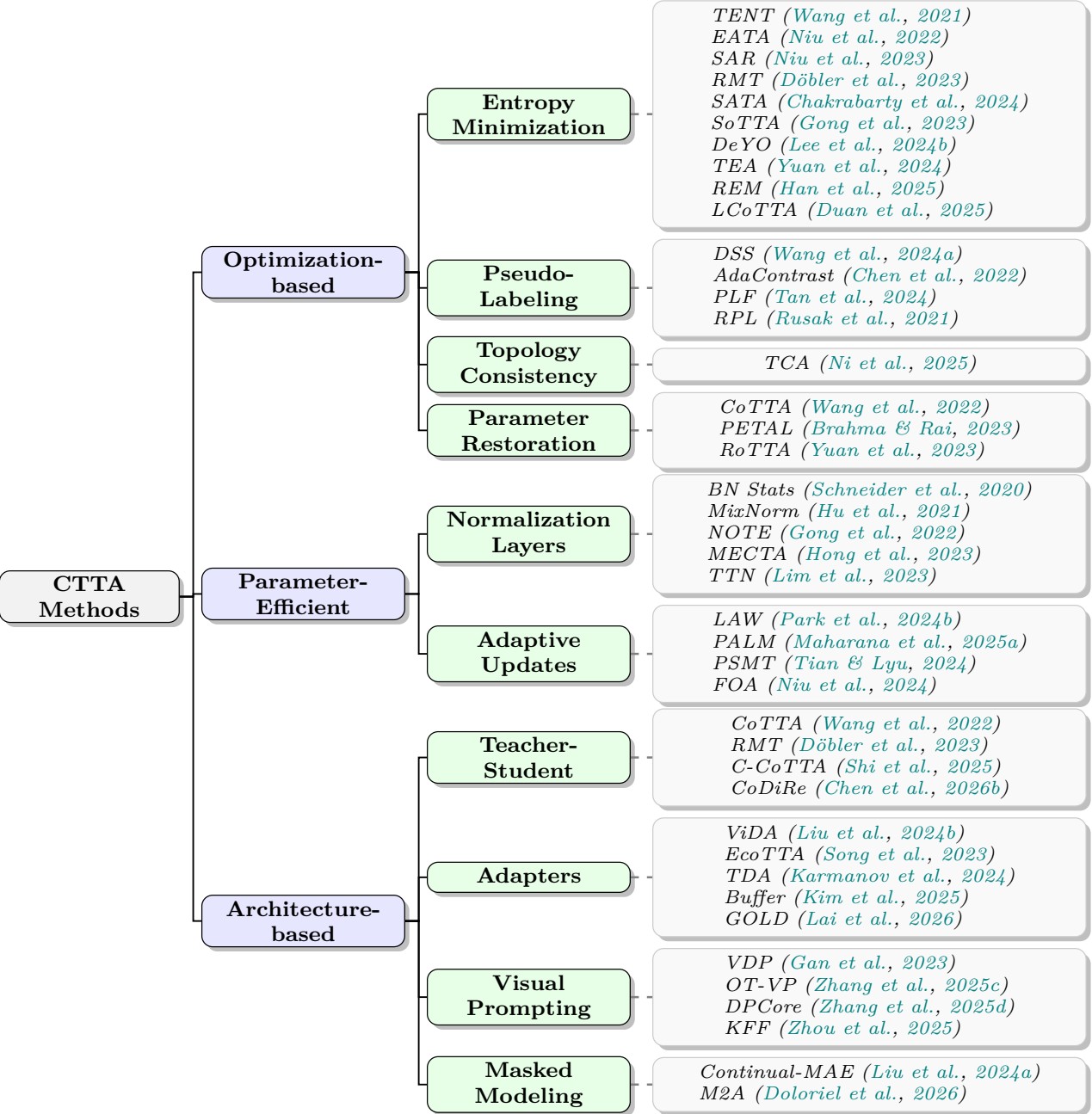

Figure 3: Hierarchical taxonomy of representative CTTA methods. Methods are organized into three main families: *Optimization-based* approaches that design self-training objectives (entropy minimization, pseudo-labeling, parameter restoration, topology preservation), *Parameter-Efficient* methods that selectively update model components (normalization layers, adaptive updates), and *Architecture-based* approaches that introduce additional modules (teacher-student, adapters, prompting, masked modeling). Years indicate the publication year. Note that some methods, including TTA methods, span multiple categories. In §4, we discuss other methods in great detail.

**Foundational TTA as CTTA Baselines.** The evolution of CTTA is deeply rooted in foundational Test-Time Adaptation (TTA) paradigms. Seminal works like TENT (Wang et al., 2021) introduced entropy minimization to update normalization layers, while TTT (Sun et al., 2020) pioneered self-supervised auxiliary tasks for online generalization. Other pivotal methods frequently cited in the CTTA landscape (Liang et al.,

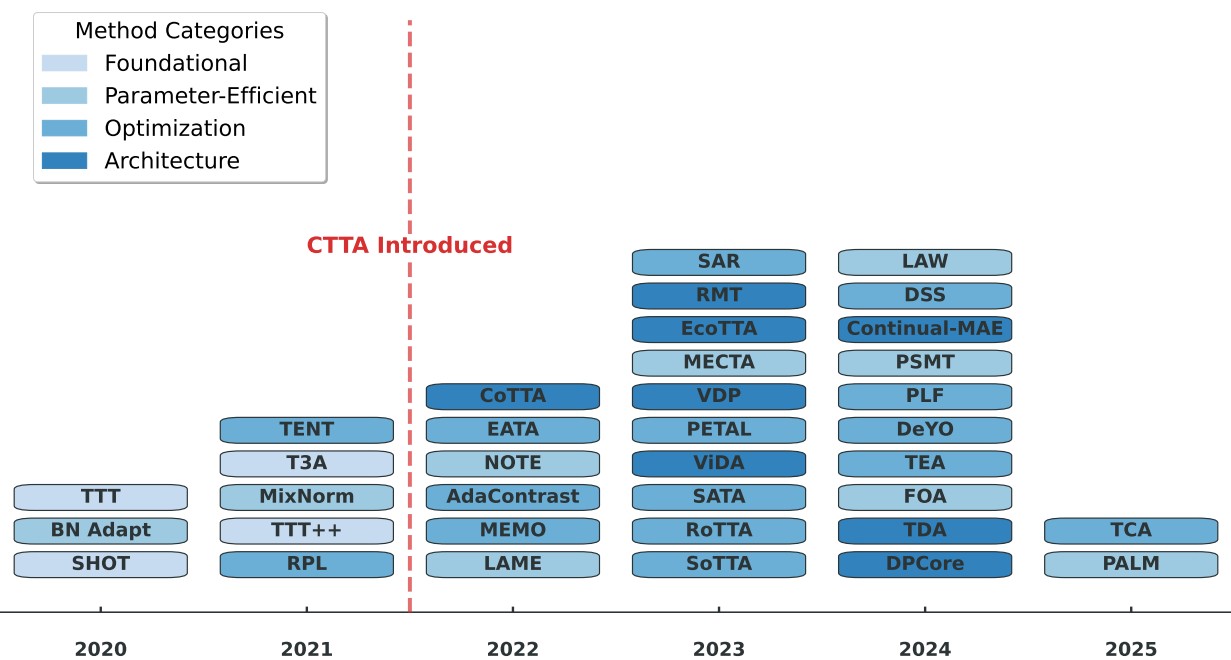

Figure 4: Timeline of methods from 2020 to 2025. The field evolved from foundational TTA works (TTT, TENT) to the introduction of Continual TTA (CTTA) by CoTTA in 2022, which sparked rapid growth in methods addressing catastrophic forgetting and error accumulation under continual distribution shifts. Methods are color-coded by their primary approach. *This includes some methods outside the paper's detailed taxonomy scope.

2025) include SHOT (Liang et al., 2020) for source-free hypothesis transfer and MEMO (Zhang et al., 2022), which leverages test-time augmentations to reduce prediction uncertainty, and AdaContrast (Chen et al., 2022), which introduces a novel way to leverage self-supervised contrastive learning to facilitate target feature learning. In the specific context of vision-language and large-scale models, recent advancements such as SAT (Mishra et al., 2024) emphasize utilizing fixed text embeddings as anchors, while BATCLIP (Maharana et al., 2025b) explores the robust adaptation of both visual and text encoders in foundational models like CLIP (Radford et al., 2021) to non-stationary distributions. These foundational strategies are often evaluated on catastrophic forgetting if the model does not reset between sequential tasks $\{\mathcal{T}_i\}_{i=1}^{K}$. As a remark, we would like to highlight that most existing TTA works can be readily extended and applied to CTTA. In this survey, we discuss TTA methods that regularly appear in CTTA discussions and baselines.

## 4.1 Optimization-based Methods

### 4.1.1 Entropy Minimization

CTTA extends the paradigm of TTA by addressing the challenges of catastrophic forgetting and error accumulation, which arise due to continual parameter updates and the propagation of noisy pseudo-labels, respectively. Methods draw inspiration from entropy minimization (Shannon, 1948), a core self-training objective initially introduced by the seminal work TENT (Wang et al., 2021) for TTA, which aims to adapt a source model to the target distribution at test-time. This does not modify the source pretraining phase. Such a loss function behaves as a proxy objective by encouraging the model to make more confident predictions on test data involving rapid change in domains. Entropy measures the uncertainty of model predictions, and the intuition of minimizing in such a setting is to nudge the model towards more decisive outputs, 'forcing' it to align better with the test data's distribution, without any label supervision. The gradient flow from the entropy loss adjusts the parameters to reduce the prediction uncertainty. The general

optimization objective is formulated as,

$$\theta^{t+1} = \arg\min_{\theta} \mathcal{H}(p_{\theta}(.|x_t)), \tag{2}$$

where $\mathcal{H}$ denotes the entropy. As a self-training objective, $\mathcal{H}$ is the Shannon entropy (Shannon, 1948) over the predictive class distribution $p_t(c)$, computed over all the classes in $\mathcal{Y}$ as,

$$\mathcal{H}(p_t) = -\sum_{c \in \mathcal{Y}} p_t(c) \log(p_t(c)). \tag{3}$$

Entropy minimization lays the foundational optimization objective (coupled with others) for most CTTA works. Depending upon the ease of extension, a few TTA works have been frequently used in the CTTA literature, too, where parameters are updated without any model reset. TENT (Wang et al., 2021) minimizes the Shannon entropy of the model predictions and updates the parameters. Inspired by TENT, many TTA approaches, and eventually CTTA approaches, extend this idea. EATA (Niu et al., 2022) adapts the parameters based on low-entropy samples, as high-entropy or uncertain predictions degrade the performance. MEMO (Zhang et al., 2022) augments each test image multiple times and updates the model parameters by minimizing the entropy of the average model predictions. STAMP (Yu et al., 2024) uses a replay buffer and minimizes a weighted entropy that places higher weights on low-entropy test samples. The buffer is updated with low-entropy samples. SAR (Niu et al., 2023) studies the failure mode of entropy minimization. It proposes encouraging the model to reside in flat regions of the entropy loss landscape to obtain good generalization and be robust to large gradients, by optimizing the following minimax entropy objective,

$$\min_{\theta^t} \max_{||\Delta_{\theta^t}||_2 \leq r} \mathcal{H}(x_t; \theta^t + \Delta_{\theta^t}). \tag{4}$$

The inner objective is to find a perturbation $\Delta_{\theta^t}$ within the Euclidean ball of radius $r$ while maximizing the entropy $\mathcal{H}$. RMT (Döbler et al., 2023) proposes using a symmetric cross-entropy loss. DeYO (Lee et al., 2024b) proposes a probability difference metric that is computed between predictions before and after augmentations, helping in sample selection and thereby entropy minimization. SATA (Chakrabarty et al., 2024) builds on the idea of self-distillation (Yuan et al., 2019; Yang et al., 2019), employing two cross-entropy losses: one between the predictions of the adapted model and the source model, and another between the predictions of the adapted model on an augmented input and the corresponding source model predictions. The guidance from the source model helps in minimizing catastrophic forgetting. SoTTA (Gong et al., 2023) addresses the challenge of noisy samples by maintaining a class-balanced memory buffer and applying entropy sharpening to filter unreliable predictions. TEA (Yuan et al., 2024) introduces test-time energy adaptation, which replaces entropy with an energy-based objective that provides more stable gradients under severe distribution shifts. The appeal of entropy minimization in CTTA stems from its ability to facilitate rapid adaptation to new target distributions by encouraging the model to produce more confident predictions. However, the continual distribution shifts pose a challenge. Relying heavily on potentially noisy pseudo-labels generated through entropy minimization can lead to a gradual accumulation of errors in the model's parameters. REM (Han et al., 2025) proposes a masked entropy minimization loss to avoid model collapse. LCoTTA (Duan et al., 2025) theoretically and empirically shows that entropy minimization can be degenerate. They show that gradients obtained via entropy minimization form a low-dimensional structure caused by entropy-truthful batch samples with highly correlated gradients. Similarly, Lee et al. (2024b) provides a theoretical analysis highlighting the limited reliability of entropy as a robust measure of confidence, especially in the context of evolving data distributions. MGP (Xiong et al., 2026) provides a geometric view and argues that there is a *manifold erosion* where spectral analysis reveals that reliable gradients lie in a stable low-rank subspace. In contrast, gradients from confident mispredictions are high-rank but consistently leak into this protected subspace. This leakage hampers the representations.

### 4.1.2 Pseudo-Labeling

Beyond entropy minimization, CTTA approaches often incorporate additional guidance through pseudo-labels (Lee et al., 2013) generated by the model itself. Intriguingly, both entropy minimization and the use

of pseudo-labels share a common objective in CTTA: to boost the confidence level of the model's predictions. This is formulated as,

$$\widehat{y}^t = \underset{\mathcal{Y}}{\arg\max}\, p(f_{\theta^t}(x_t)), \tag{5}$$

where $\widehat{y}^t$ is the predicted label, $f_{\theta^t}(x_t)$ denotes the model prediction on input batch $x_t$, parameterized by $\theta^t$ at time-step $t$ over the label space $\mathcal{Y}$, and $p(\cdot)$ denotes the softmax probability over label space $\mathcal{Y}$. It can be seen that pseudo-labels can be inaccurate due to continual shifts in distribution. So, CTTA approaches focus on refining the pseudo-labels' quality to provide more enhanced supervision. DSS (Wang et al., 2024a) is built upon an important observation. Since pseudo-labels are noisy and not trustworthy, and yet most models rely on them to update parameters through a self-training loss, it proposes a dynamic thresholding technique to filter out high-quality pseudo-labels from the low-quality ones. AdaContrast (Chen et al., 2022) utilizes a self-supervised contrastive learning framework. Pseudo-labels are refined by leveraging the knowledge from the test sample's neighborhood. RPL (Rusak et al., 2021) proposes robust pseudo-labeling by exploring hard and soft pseudo-labeling. For robust pseudo-labeling, they recommend using a generalized cross-entropy loss. PLF (Tan et al., 2024) proposes a self-adaptive thresholding mechanism to filter out reliable pseudo-labels in a test batch. The initial thresholds are gradually matched to the confidence of the model prediction. DAS (Zhu et al., 2026) constructs a synthetic knowledge base offline and dynamically bridges it to match the target domain at input, statistical, and representation levels, using the adapted proxies with ground-truth labels as reliable supervision signals through proxy-based cross-entropy, contrastive, and self-training losses. This is fundamentally a loss design innovation for addressing unreliable pseudo-label supervision.

*Discussions.* The key similarity between pseudo-labeling and entropy minimization lies in error accumulation due to continual shifts in distributions. While efforts to generate more confident pseudo-labels aim to improve the supervision of the continually adapting model, a crucial limitation remains: there is no assurance that either pseudo-labels or entropy minimization will remain dependable when faced with severe distribution shifts.

### 4.1.3 Topological Consistency

Standard entropy minimization and pseudo-labeling methods often suffer from feature space collapse, where class clusters merge or distort significantly under continual shifts. TCA (Topological Consistency Adaptation) (Ni et al., 2025) addresses this by enforcing a *class topological consistency constraint.* Instead of adapting samples in isolation, TCA explicitly maintains the geometric relationships between class centroids. It minimizes the distortion of inter-class distances and enforces intra-class compactness, ensuring that the semantic structure of the feature space remains stable even as the domain shifts continually. This geometric regularization acts as a critical counter-balance to the 'entropy bias' where models become overconfident in incorrect predictions.

### 4.1.4 Parameter Restoration

To minimize error accumulation due to noisy labels, certain model parameters of the adapted model are restored to the source model parameters. This ensures stability during continual adaptation by partially or periodically reverting model parameters to a more reliable state, helping mitigate the risk of overfitting (Wang et al., 2022). Given the flattened source parameters $\widehat{\theta}^S$ and the *adapted* model weights $\widehat{\theta}^{t+1}$ at time-step $t+1$, the restoration is done as,

$$\mathbf{m} \sim \text{Bernoulli(p)} \tag{6}$$

$$\widehat{\theta}^{t+1} = \mathbf{m} \odot \widehat{\theta}^S + (1 - \mathbf{m}) \odot \widehat{\theta}^{t+1}, \tag{7}$$

where $p$ is a stochastic probability and $\mathbf{m} \in \{0, 1\}^{|\theta|}$ is a binary mask to control the set of parameters in $\widehat{\theta}^{t+1}$ to be restored. CoTTA (Wang et al., 2022) applies the above mechanism uniformly across all parameters by reverting them to the source model weights. Notably, this approach does not account for the individual behavior or importance of each parameter, treating all parameters equally during restoration. In PETAL (Brahma & Rai, 2023), the Fisher Information Matrix (FIM) (Sagun et al., 2017), computed as the diagonal of the product of the gradient and its transpose, is used to measure the parameter importance. This diagonal

estimate serves as a proxy for how crucial each parameter is to the model's predictive behavior. $\mathbf{m}$ is set to 1 if the diagonal FIM is below a predefined threshold, else it is set to 0. This mechanism enables targeted, uncertainty-aware updates while maintaining stability during CTTA. RoTTA (Yuan et al., 2023) extends parameter restoration with a tiered memory bank that maintains category-balanced samples and applies robust batch normalization to stabilize adaptation under temporally correlated test streams.

*Discussions.* The reliable gains in this family come not from any single self-training objective but from coupling one with an explicit anti-forgetting mechanism: entropy or cross-entropy losses paired with stochastic source restoration (Wang et al., 2022), Fisher-weighted restoration (Brahma & Rai, 2023), or category-balanced replay (Yuan et al., 2023), while filtering the update signal via low-entropy selection (Niu et al., 2022; Gong et al., 2023), augmentation-based agreement (Lee et al., 2024b), or symmetric losses (Döbler et al., 2023) recover most of the remaining gap. The fragility is structural: the confidence signal these methods optimize is itself unreliable under severe shift, and recent analyses (Lee et al., 2024b; Duan et al., 2025) show that entropy minimization can collapse to a degenerate low-dimensional solution whenever the batch is dominated by entropy-truthful samples.

## 4.2 Parameter-Efficient Methods

Source models have a sheer number of parameters. Directly adjusting the parameters, at test-time, can be computationally demanding, as it requires significant computational resources and gradient flow all the way back to the shallow layers. The major drawback of full fine-tuning is the loss of source knowledge, hinting at catastrophic forgetting.

### 4.2.1 Normalization Layers

An initial set of works focuses on modulating the statistics of the normalization layers, especially the batch normalization layers (BatchNorm) (Ioffe & Szegedy, 2015). BatchNorm plays a crucial role in stabilizing the training of neural networks. Its effectiveness stems from normalizing the activations within each layer by estimating their mean and variance, typically across the entire training dataset or within sufficiently large mini-batches. By maintaining more consistent activation distributions, BatchNorm significantly reduces the likelihood of encountering vanishing or exploding gradients.

BatchNorm layers maintain two distinct types of update mechanisms. The first are running statistics ($\mu$ and $\sigma^2$), which are not learned parameters. They carry no gradients and are never updated via backpropagation. Instead, they are maintained as exponential moving averages (EMA) across test batches:

$$\mu \leftarrow (1-\lambda)\,\mu + \lambda\,\mu_t, \tag{8}$$

$$\sigma^2 \leftarrow (1-\lambda)\,\sigma^2 + \lambda\,\sigma_t^2, \tag{9}$$

where $\mu_t$ and $\sigma_t^2$ are the mean and variance of the current test batch, and $\lambda$ is the EMA coefficient. The second are affine parameters $\gamma$ and $\beta$, which are the only trainable parameters in a BatchNorm layer. Together, they produce the following normalized output at time-step $t$: given the input features $g_t$ to a BatchNorm layer, the transformed features are,

$$\widehat{g}_t = \gamma \odot \frac{g_t - \mu_t}{\sqrt{\sigma_t^2}} + \beta, \tag{10}$$

where $\mu_t$ and $\sigma_t^2$ denote the mean and variance of the test batch, respectively. We provide an illustration in Figure 5.

Building on this, TENT (Wang et al., 2021) proposes adaptation (also for continual) via entropy minimization, exploiting both update mechanisms. First, the stored source statistics ($\mu_S$, $\sigma_S^2$) are replaced with batch-level statistics of the target distribution. Li et al. (2017) suggest that the target domain's information can be conveyed through the statistical interpretation of the BatchNorm parameters. Second, the affine parameters ($\gamma$ and $\beta$), which constitute <1% of the parameters, are trainable and updated based on their gradients of the Shannon entropy $\mathcal{H}$ (Shannon, 1948) of the model's predictions on the target batch at

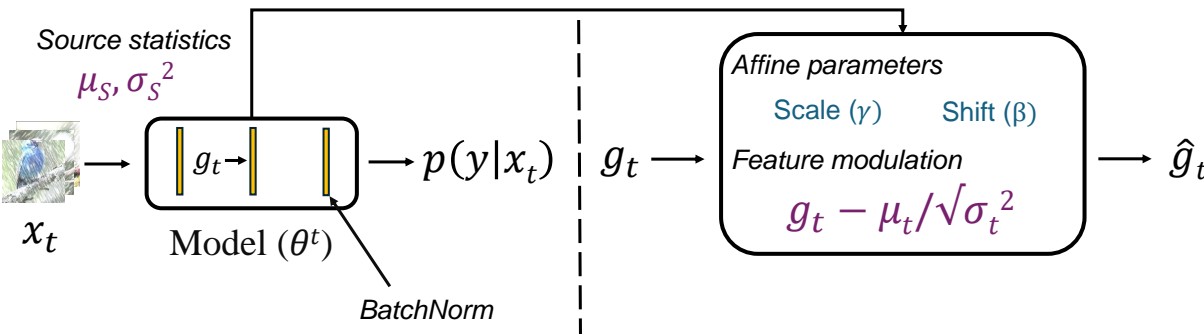

Figure 5: Illustration of estimating BatchNorm statistics and feature modulation at test-time. For an input feature $g_t$, the transformed output $\hat{g}_t$ is computed using the batch-wise mean $\mu_t$ and variance $\sigma_t^2$ from the target distribution: $\hat{g}_t = \gamma \odot \frac{g_t - \mu_t}{\sqrt{\sigma_t^2}} + \beta$, where $\gamma$ and $\beta$ are learnable affine parameters updated during adaptation.

time-step $t$ as,

$$\gamma^{t+1} \leftarrow \gamma^t + \nabla_{\gamma^t} \mathcal{H}(.), \tag{11}$$

$$\beta^{t+1} \leftarrow \beta^t + \nabla_{\beta^t} \mathcal{H}(.). \tag{12}$$

In the presence of noisy test samples, the performance of TENT is impacted. Continual updates of affine parameters lead to overfitting on future test tasks. To stabilize the adaptation process, increasing the batch size is an alternative. As in TENT and other works updating norm statistics (Hu et al., 2021; Wang et al., 2021), a common choice of batch size is 200 on CIFAR-C datasets. Such a practice falters with GPU memory constraints when batches can be of a smaller size. Additionally, relying on the batch size, especially at test-time is an unrealistic assumption. To work with any batch size in an online manner, MixNorm (Hu et al., 2021) predicts norm statistics for each test sample and then refines these estimates using a moving average of global statistics accumulated from previous test samples. Given the input image and its augmentation, let the input feature and the corresponding augmented feature be $g_t$ and $g_t'$, where $g_t$ and $g_t' \in \mathbb{R}^D$. The mean and variance of the features would be,

$$\mu = \frac{\sum_{i=0}^{Q} \sum_{j=0}^{R} g_t[i,j]}{QR}, \tag{13}$$

$$\sigma^2 = \frac{\sum_{i=0}^{Q} \sum_{j=0}^{R} (g_t[i,j] - \mu)^2}{QR}, \tag{14}$$

where $Q$ and $R$ denote the first two dimensions of the feature map, usually the batch size and channel dimensions, respectively. The local statistics, $\mu_{\text{local}}$ and $\sigma_{\text{local}}^2$, are computed as,

$$\mu_{\text{local}} = \frac{\sum_{k \in \{g_t, g_t'\}} \sum_{i=0}^{Q} \sum_{j=0}^{R} k[i,j]}{2QR}, \tag{15}$$

$$\sigma_{\text{local}}^2 = \frac{\sum_{k \in \{g_t, g_t'\}} \sum_{i=0}^{Q} \sum_{j=0}^{R} (k[i,j] - \mu_{\text{local}})^2}{2QR}. \tag{16}$$

Let $\mu_{t-1}$ and $\sigma_{t-1}^2$ be the running or global statistics up to the current time-step $t$, computed as a moving average with hyperparameter $\tau$. The normalized input features at time-step $t$ with affine parameters $\gamma$ and $\beta$ (from the source model) are

$$\hat{g}_t = \gamma \odot \frac{g_t - \mu_{\text{mixed}}}{\sqrt{\sigma_{\text{mixed}}^2}} + \beta, \tag{17}$$

$$\hat{g}'_t = \gamma \odot \frac{g'_t - \mu_{\text{mixed}}}{\sqrt{\sigma^2_{\text{mixed}}}} + \beta, \tag{18}$$

where $\mu_{\text{mixed}}$ and $\sigma^2_{\text{mixed}}$ are given by

$$\mu_{\text{mixed}} = \lambda\mu_{\text{local}} + (1 - \lambda)\mu_{t-1}, \tag{19}$$

$$\sigma^2_{\text{mixed}} = \lambda\sigma^2_{\text{local}} + (1 - \lambda)\sigma^2_{t-1}. \tag{20}$$

However, the momentum factor $\lambda$ to compute the moving average is fixed and is independent of the distribution. Mirza et al. (2022) push this by proposing a dynamic way to determine the momentum, based on a decay factor. ERSK (Niloy et al., 2024) uses the KL divergence of norm statistics between the current test batch and the source model to determine the momentum. Though fancy, moving averages can affect the optimization of gradients and normalization when updating the affine parameters of BatchNorm layers. NOTE (Gong et al., 2022) takes inspiration from Instance Normalization (Ulyanov et al., 2016) and computes instance-wise statistics to eliminate reliance on correlated data in the batch. To leverage the source norm statistics $\mu_S$ and $\sigma^2_S$, BN Stats Adapt (Schneider et al., 2020) uses a weighted sum of the source and target statistics. When the number of test samples is small, the effective mean and variance of the BatchNorm layers are computed as, $\widehat{\mu} = \frac{N_0}{N_0 + n_t}\mu_S + \frac{n_t}{N_0 + n_t}\mu_t$, $\widehat{\sigma^2} = \frac{N_0}{N_0 + n_t}\sigma^2_S + \frac{n_t}{N_0 + n_t}\sigma^2_t$. $N_0$ is a hyperparameter to balance the trade-off between source and test statistics, and $n_t$ is the number of test samples. Selection of $N_0$, as a hyperparameter, is a limitation. TTN (Lim et al., 2023) proposes domain-shift-aware batch normalization that dynamically adjusts the weight of BatchNorm layer updates based on the estimated severity of domain shift for each layer.

*Limitations.* Since BatchNorm operates at a batch level, the activations are shifted and scaled, which lack inter-feature correlations (Huang et al., 2018). While adaptation methods have been proposed to overcome this, there are potential limitations of norm updates at test-time. Specifically, such updates often assume that distributional changes can be captured through first- and second-order moments, which may be insufficient when the shift affects more complex feature dependencies or semantic content. As discussed, small batch sizes harm the estimation error of $\mu_t$ and $\sigma^2_t$ statistics, making them noisy and unreliable. Moving average-based (Hu et al., 2021; Niloy et al., 2024; Mirza et al., 2022) methods address this, as discussed, but norm statistics alone may not offer sufficient expressiveness in continual test-time scenarios. Their drawbacks can be seen in the results and motivation of CTTA papers.

*BatchNorm alternatives.* To reduce high GPU memory consumption and make CTTA approaches suitable for edge devices, MECTA (Hong et al., 2023) proposed MECTA Norm, a substitute for BatchNorm. This modification is particularly beneficial in scenarios where the computational overhead associated with large batch sizes, high-dimensional feature channels, and the need to update numerous layers would otherwise be very difficult. EcoTTA (Song et al., 2023) proposed meta networks to be attached to each block of the source model during adaptation. The meta network comprised a 'conv block' followed by a BatchNorm layer. Such a setup resulted in a memory-efficient CTTA framework.

*Discussions.* Adjustment of the normalization statistics in BatchNorm layers has served as the cornerstone of efficient TTA and, thereby, CTTA approaches. However, it is not model-agnostic. Modern computer vision models like Vision Transformers (Dosovitskiy et al., 2021) have garnered attention, which have LayerNorm layers and do not benefit. LayerNorm normalizes across features within a single sample rather than across the batch, making it insensitive to batch-level distribution shifts and thus incompatible with BatchNorm-centric TTA methods. So, there is a need for model-agnostic adaptation methods. Recent works have started to explore alternatives, such as using model uncertainty to guide adaptation (Maharana et al., 2025a), or introducing lightweight adapters that are normalization-independent (Liu et al., 2024b).

### 4.2.2 Adaptive Parameter Updates

Memory- and compute-intensive fine-tuning poses a challenge for CTTA, especially in resource-constrained environments. To address this, a new line of CTTA research draws inspiration from the heterogeneous transferability of pre-trained layers (Chatterji et al., 2019; Neyshabur et al., 2020). These works observe that not all layers in a deep model contribute equally to generalization across distributions. Building on

this, Lee et al. (2023b) proposes that, depending on the nature of the target distribution, a carefully selected subset of source model layers can be heuristically fine-tuned, avoiding the overhead of full-model updates. This naturally raises an important question: how can we automatically and efficiently determine which layers to adapt for a given shift? This layer selection problem is non-trivial in CTTA, where supervision is absent and distribution shifts are dynamic and rapidly changing.

LAW (Park et al., 2024b) computes a layer-wise importance score based on FIM, which serves as an approximation of the Hessian matrix of the log-likelihood (Sagun et al., 2017). For the $j^{\text{th}}$ parameter of a layer, from the model parameters $\theta^t$ at time-step $t$, the score is computed as,

$$h(\theta_j^t, x_t) = \nabla_{\theta_j^t} \log(p(f_{\theta^t}(x_t))), \tag{21}$$

where the gradient is computed over the log softmax probability $p(\cdot)$ of the model predictions $f_{\theta^t}(x_t)$. The FIM is,

$$I_j^t = \mathbb{E}_{x_t}[h(\theta_j^t, x_t)h(\theta_j^t, x_t)^T]. \tag{22}$$

Eqns. 21 and 22 are computed for every parameter of the continually fine-tuned model. To capture domain-level information from past batches, additionally, Eqn. 22 is computed as a running summation ($\widehat{I_j^t}$) with the final 'adapted' learning rate weight being the square root of the trace of $\widehat{I_j^t}$ of the $j^{\text{th}}$ parameter. This way, each parameter is adaptively updated with a different step size depending on the domain. To address the limitations of relying on noisy pseudo-labels for estimating layer importance (as in Eqn. 21), PALM (Maharana et al., 2025a) introduces an uncertainty-guided mechanism for automatic layer selection during CTTA. The core idea is to leverage model prediction uncertainty as a proxy for identifying which layers are most sensitive to distributional shifts and therefore should be adapted. Specifically, PALM computes the gradient of the KL divergence between a smoothed model output and a uniform distribution, which captures the degree of predictive uncertainty. For the $j^{\text{th}}$ parameter of the $k^{\text{th}}$ layer, the gradient is given by,

$$\nabla_{\theta_{j,k}^t} \text{KL}\left(\text{softmax}\left(\frac{f_{\theta^t}(x_t)}{\tau}\right) \,\Big\|\, u\right), \tag{23}$$

where $\tau$ is the temperature parameter and $u$ is a uniform distribution over classes. An $L_1$ norm on Eqn. 23 reveals important possible intuitions. Layers with smaller aggregated gradients are considered more relevant for adaptation under shift, since they are more uncertain about the domain, and are thus selected for fine-tuning. PALM freezes the remaining model parameters to ensure minimal forgetting of source knowledge. In line with prior work on parameter importance estimation (Molchanov et al., 2019), PALM maintains a running average of sensitivity scores $s_{j,k}^t$ to capture historical domain-level information. The smoothed sensitivity score $\widehat{s_{j,k}^t}$ for each parameter is updated using an EMA,

$$\widehat{s_{j,k}^t} = \lambda s_{j,k}^t + (1-\lambda)\widehat{s_{j,k}^{t-1}}, \tag{24}$$

where $\lambda$ is a smoothing factor $\in [0,1]$ to control the influence of previous sensitivities. The adaptation rate for each selected parameter is then modulated based on the uncertainty in its sensitivity, defined as,

$$v_{j,k}^t = \frac{|s_{j,k}^t - \widehat{s_{j,k}^t}|}{\widehat{s_{j,k}^t} + \epsilon}, \tag{25}$$

where $\epsilon$ is to avoid division by 0. This formulation assigns higher learning rates to parameters with both low sensitivity and high temporal variability, encouraging localized adaptation to new domains while preserving the generalization learned from source data. Overall, PALM provides a principled approach for layer-wise selective adaptation, balancing plasticity and stability in a continual test-time setting. One of PALM's key advantages lies in its ability to freeze a substantial portion of the model parameters, with the majority of the selected parameters, i.e., ∼63%, corresponding to the affine parameters of BatchNorm layers (Wang et al., 2021; Ioffe & Szegedy, 2015). FOA (Niu et al., 2024) takes a different approach by enabling test-time adaptation through forward passes only, eliminating the need for backpropagation. This is achieved by optimizing learnable input prompts using covariance adaptation, making it particularly suitable for resource-constrained deployment. For a teacher-student network, PSMT (Tian & Lyu, 2024) selectively decides the

subset of teacher parameters to update. Inspired by Liu et al. (2021), that only a subset of parameters are effective in an over-parameterized network, PSMT proposes selective knowledge distillation between the models. To regulate updates to the student, PSMT introduces a quadratic regularization constraint weighted by the diagonal Fisher Information Matrix (FIM) (Sagun et al., 2017), which captures the sensitivity of each parameter. This ensures that updates avoid altering parameters critical to the model's generalization. For the teacher model, PSMT identifies important parameters via a threshold on the diagonal FIM. Only parameters with sensitivity exceeding this threshold are updated. This constrains it from drifting under domain shift and reduces the risk of forgetting.

*Discussions.* There have been limited CTTA works on automatically selecting a subset of layers and adaptively deciding the degree of fine-tuning on a continual stream of tasks. However, they're tricky to work with at test-time. Adaptive learning rate schemes rely on gradient-based signals, which can be highly noisy without supervision. Noisy gradients can cause erratic or unstable updates. These methods also introduce extra hyperparameters, which could be non-trivial to tune at test time. Poor hyperparameter choices can amplify sensitivity noise or make the model too rigid.

### 4.3 Architecture-based Methods

In this section, we discuss CTTA methods that attach additional parameters to the source model for efficient continual adaptation to different distributional shifts. This includes memory-intensive architectures like teacher-student, lightweight modules, or masked tokens, which are updated at test-time.

#### 4.3.1 Teacher-Student

In the context of semi-supervised learning, teacher–student frameworks such as the Mean Teacher model (Tarvainen & Valpola, 2017) have been extensively studied. To enhance the stability and quality of the teacher's predictions, the teacher model is formed by taking an exponential moving average (EMA) (Klinker, 2011) of the student model's parameters over training steps. A consistency regularization is then applied between the outputs of the student and teacher models, encouraging the student to produce stable predictions for different augmentations while gradually improving through supervision from the teacher.

CoTTA (Wang et al., 2022) proposed a similar framework, as illustrated in Figure 6, where at $t=0$, both networks are initialized to $\theta^S$. Inspired by (Tarvainen & Valpola, 2017; Polyak & Juditsky, 1992), the teacher parameters $\theta_{\text{tea}}^t$ are updated using an EMA of the student parameters,

$$\theta_{\text{tea}}^{t+1} \leftarrow \lambda\theta_{\text{tea}}^t + (1-\lambda)\theta_{\text{stu}}^{t+1}, \tag{26}$$

where $\lambda$ is the momentum coefficient (typically set to 0.999), and $\theta_{\text{stu}}^{t+1}$ denotes the updated student parameters at time-step $t+1$. To update the student model, the loss is a cross-entropy between student and teacher predictions as,

$$\mathcal{L}_{CE} = -\sum_{c\in\mathcal{Y}} p_{\text{tea}}(c|x_t)\log(p_{\text{stu}}(c|x_t)), \tag{27}$$

with $p_{\text{tea}}$ and $p_{\text{stu}}$ being the teacher and student predictive distributions for class $c$, respectively. This serves as a consistency loss. If the source model's confidence in a given sample is low, CoTTA uses the average of multiple teacher predictions under hard test-time augmentations to generate a more reliable target. Otherwise, it uses the original teacher's prediction. This strategy helps mitigate confirmation bias from uncertain predictions, ensuring that the student learns from stable, high-confidence guidance. RMT (Döbler et al., 2023) extends the teacher-student framework by addressing a critical limitation: the susceptibility of standard mean teacher updates to noisy or corrupted test samples in continual streams. While CoTTA uses a standard EMA to update the teacher, RMT proposes a robust mean teacher update mechanism that filters unreliable samples before incorporating them into the teacher's knowledge. Specifically, RMT maintains a memory bank of test samples and employs a symmetric cross-entropy loss that is less sensitive to noisy pseudo-labels compared to standard cross-entropy. Another approach, by the name of ViDA (Liu et al., 2024b), uses a consistency loss between the student and teacher predictions. The teacher receives strongly augmented views of the input batch, while the student processes the original test batch. This design encourages the student to learn stable representations by aligning with the teacher's predictions, behaving as soft pseudo-labels. TDA

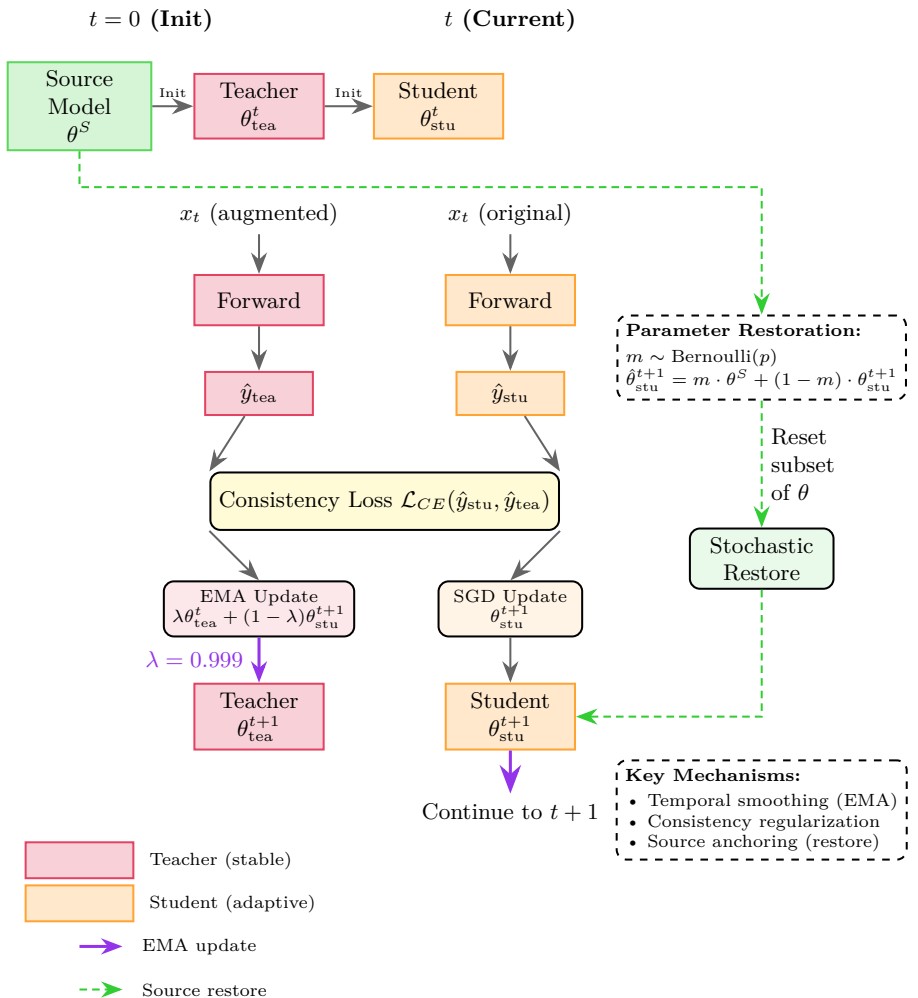

Figure 6: Teacher-Student Framework: The teacher model is updated via an exponential moving average (EMA) of student weights, providing stable pseudo-labels. Stochastic parameter restoration periodically resets a subset of student parameters to source weights to prevent catastrophic forgetting.

(Karmanov et al., 2024) introduces a training-free dynamic adapter specifically designed for vision-language models. By combining positive and negative caches based on prediction confidence, TDA achieves efficient adaptation without requiring gradient updates, significantly reducing computational overhead compared to prior methods. VDP (Gan et al., 2023) employs a cross-entropy loss between the student and teacher predictions, along with a regularization term that penalizes parameters sensitive to domain shifts. This regularization is implemented as a weighted $L_2$ distance between a subset of model parameters from the current and previous batches. It encourages the model to update only shift-resilient parameters, thereby preserving domain-invariant representations while enabling effective domain-specific learning. EMA updates the teacher model parameters. DPCore (Zhang et al., 2025d) addresses the limitation of fixed prompts by maintaining a dynamic prompt coreset that evolves with the changing target distribution. The coreset is updated based on sample representativeness and prediction confidence, enabling more robust adaptation across diverse domain sequences. C-CoTTA (Shi et al., 2025) introduces controllable continual test-time adaptation by explicitly preventing category encroachment during adaptation, thereby maintaining clearer decision boundaries between classes under domain shift. Most methods update all the teacher parameters by EMA, which could lead to forgetting and error accumulation. PSMT (Tian & Lyu, 2024) selectively decides the subset of teacher parameters to update as discussed in 4.2.2. CoDiRe (Chen et al., 2026b) uses a frozen

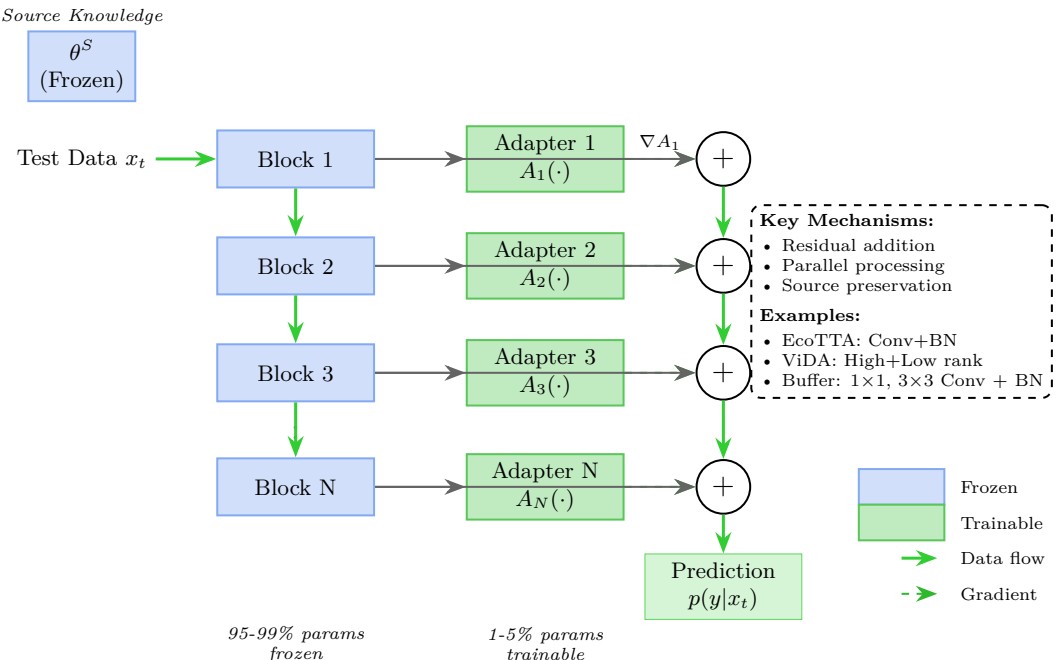

Figure 7: Adapters: Parallel branch architecture for CTTA. Lightweight adapter modules are inserted along-side frozen source model blocks, with outputs combined via residual connections. Only 1-5% of parameters are trainable (adapters), while 95-99% remain frozen (source blocks), enabling efficient adaptation while preserving source knowledge. Examples include EcoTTA (Song et al., 2023) (convolutional adapters with BatchNorm), ViDA (Liu et al., 2024b) (high-rank and low-rank adapters) and Buffer (Kim et al., 2025).

CLIP vision-language model as an external teacher to provide reliable supervisory signals via distillation loss. This sidesteps unreliable pseudo-label and self-referential feedback loops in standard CTTA.

*Discussions.* While teacher-student models have been extensively used in the CTTA literature, they come with certain limitations. 1) Rapid distributional shifts of high severity could lead to the teacher model being wrong, leading the student to reinforce the errors. 2) CTTA demands that the model adapt to new domains without forgetting past knowledge or overfitting to transient noise. As seen, most frameworks lack mechanisms to control forgetting or rebalancing domain-agnostic vs. domain-specific knowledge. 3) And the most important is high computational overload, where maintaining two networks increases memory usage and computational cost, which may not be ideal for real-time or resource-constrained deployment settings.

### 4.3.2 Adapters

To address the memory concerns of CTTA, EcoTTA (Song et al., 2023) attaches meta-networks to the source model, i.e., to each block of the model; lightweight Conv blocks and BatchNorm layers are attached. During adaptation, the entire source model remains frozen, ensuring that the original parameters are preserved and avoiding catastrophic forgetting. Only the meta-networks are updated, making the adaptation process computationally efficient and memory-light. To adapt, a confidence-based entropy as in Eqn. 3 is used along with an $L_1$ regularization that bridges the output of the proposed meta-network and the source feature extracted between each block of the source model. EcoTTA achieves a favorable trade-off between adaptability, robustness, and computational cost. In ViDA (Liu et al., 2024b), high-rank and low-rank adapters are attached to the linear/conv layers of the source model. The rationale for such an architectural design is as follows: the high-rank adapter has a stronger representational capacity and is thereby better suited to capture stable, domain-agnostic knowledge shared across continual test distributions. A high rank encodes rich, persistent representations. On the other hand, the low-rank adapter is more suited to general domain information and is limited in capacity. This enables fast adaptation without overfitting to any single

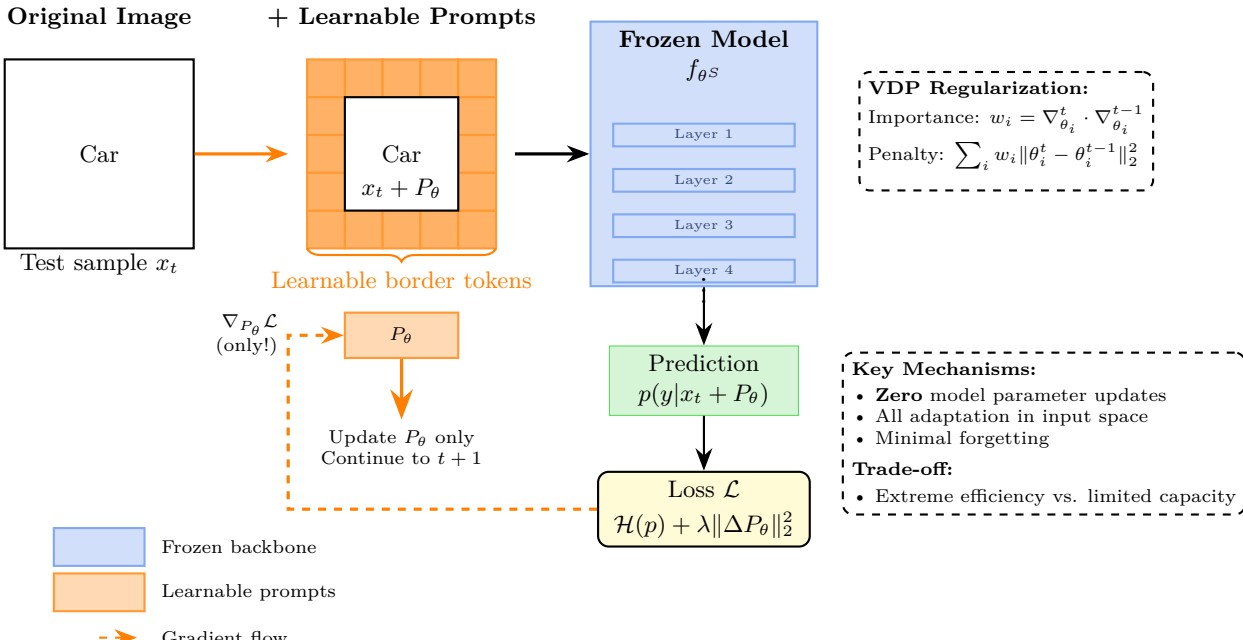

Figure 8: Visual Prompting: Input space adaptation for CTTA. The model backbone remains completely frozen while only lightweight prompt tokens (border pixels) are adapted, achieving extreme parameter efficiency at 0.1% trainable parameters.

domain and minimizes catastrophic forgetting. The design choices were also empirically verified by the authors. To dynamically balance their contributions, ViDA leverages MC Dropout (Gal & Ghahramani, 2016) to estimate prediction uncertainty at the output. When uncertainty is high, greater emphasis is placed on the high-rank adapter, whereas lower uncertainty shifts focus toward the low-rank adapter. Taking a different approach, TDA (Karmanov et al., 2024) introduces a training-free dynamic adapter specifically designed for vision-language models in the CTTA setting. Unlike EcoTTA and ViDA, which require gradient-based updates, TDA maintains positive (stores high-confidence predictions that align with the model's original knowledge) and negative caches (identifies and suppresses unreliable predictions) based on prediction confidence and uses these caches to dynamically adjust model outputs without any parameter updates. This cache-based mechanism enables efficient adaptation for large-scale vision-language models like CLIP (Radford et al., 2021), where traditional gradient-based adaptation would be prohibitively expensive. Buffer (Kim et al., 2025) proposes the insertion of a lightweight convolutional module in the early blocks of a frozen source model to address the limitations of adapting normalization layers in low batch-size settings. PAID (Wang et al., 2025c) introduces learnable orthogonal matrices, as opposed to a standard adapter, to preserve pairwise angular distances between source weights. During adaptation, only the magnitudes and the orthogonal matrices are changed. GOLD (Lai et al., 2026) uses a lightweight residual adapter that projects features onto a dynamically maintained golden subspace and learns a compact scaling vector. In Figure 7, we provide an illustrative overview.

*Discussions.* For the methods discussed above, the adapters are pre-trained on their specific source datasets. Such a setup does not provide a generalized solution to real-world deployment. Pre-trained adapters may struggle to generalize, leading to suboptimal or unstable adaptation. Ideally, CTTA strategies should enable on-the-fly, data-driven adaptation that remains agnostic to prior training environments

### 4.3.3 Visual Prompting

Taking strong inspiration from the fact that model adaptation based on noisy pseudo-labels could be inaccurate in a continual setup, VDP (Gan et al., 2023) proposes to encode domain knowledge via visual prompting (Bar et al., 2022), while keeping all source model parameters frozen. This approach is highly parameter-

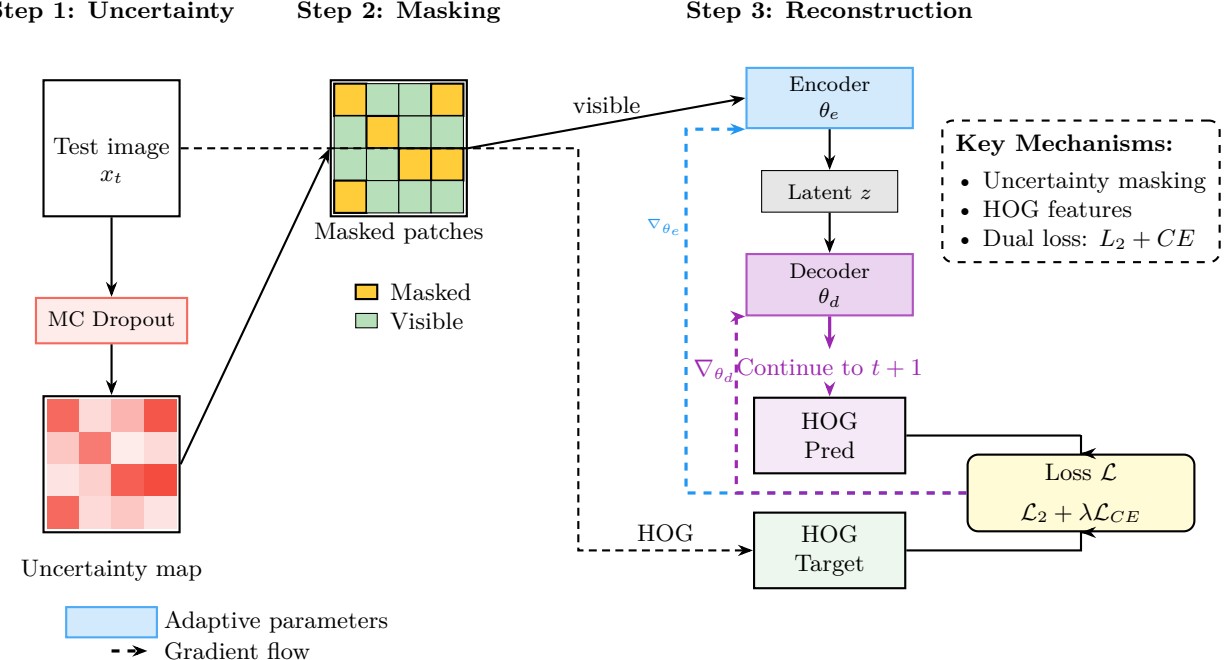

Figure 9: Masked Modeling: Uncertainty-guided reconstruction for CTTA.

efficient, as it avoids modifying the backbone and instead injects lightweight image tokens (prompts) directly into the input space. To prevent over-adaptation to a domain and to learn domain-agnostic knowledge, a weight importance factor is computed as a product of gradients between two adjacent time-steps. This is summed over evolving test distributions, weighted by the difference of student parameters, until the current distribution $\mathcal{T}_i$. Such a factor serves as a parameter importance that is used with an $L_2$ regularization of model parameters from the previous time-step and domain. For domain-specific knowledge, a cross-entropy loss between the student-teacher predictions is computed. In Figure 8, we illustrate this idea. Inspired by visual prompt tuning (Jia et al., 2022), DPCore (Zhang et al., 2025d) addresses the challenge of adapting to continually evolving distributions by maintaining a dynamic prompt coreset that evolves alongside the changing target domains. DPCore selectively updates its prompt coreset based on two key criteria: sample representativeness and prediction confidence. Specifically, DPCore computes a coreset selection score that balances diversity (ensuring the prompts cover the input distribution) and reliability (filtering out low-confidence samples that may introduce noise). The coreset is incrementally updated as new test data arrives, with prompts corresponding to outdated or unreliable samples being replaced. KFF (Zhou et al., 2025) employs a class-aware fission and fusion module with visual prompts. The fission module adaptively isolates new domain knowledge into class-aware prompts to prevent interference from distinct historical data, while the fusion module employs a greedy merging strategy to efficiently integrate this new knowledge into the existing pool without prohibitive overhead.

*Discussions.* Visual prompting (Xiao et al., 2025) is a very parameter-efficient approach for CTTA. However, under large covariate shifts and severity at test-time, they may lack expressiveness. In addition to that, approaches like VDP are very sensitive to the regularization. Prompts can over-adapt to a single domain and hurt future adaptation.

### 4.3.4 Masked Modeling

Continual-MAE (Liu et al., 2024a) brings a fresh perspective to CTTA by taking inspiration from masked image modeling (He et al., 2022; Xie et al., 2022). The model identifies uncertain image patches using Monte Carlo (MC) Dropout (Gal & Ghahramani, 2016), which estimates predictive uncertainty by measuring output variance across stochastic forward passes. Based on these estimates, the most uncertain patches are masked,

and the model is trained to reconstruct their Histogram of Oriented Gradients (HOG) (Dalal & Triggs, 2005) features, rather than raw pixels, serving as a robust, handcrafted supervision signal. This mitigates the risk of overfitting to noisy or corrupted inputs, particularly in challenging domains. To further stabilize masked token learning, Continual-MAE combines this $L_2$ reconstruction loss with a cross-entropy loss that aligns predictions of a linear layer from masked views with those from the original image (see Figure 9). M2A (Doloriel et al., 2026) systematically isolates the masking family from the selection strategy, showing that spatial patch masking yields the most stable long-term adaptation.

*Discussions.* By relying on hand-engineered features like HOG for reconstruction, Continual-MAE inherently limits the expressiveness of its adaptation objective. The method assumes that local structural information captured by HOG is preserved across domains; however, under severe or high-level domain shifts, reconstructing such low-level features may not provide sufficient semantic guidance for effective adaptation. Additionally, the use of MC Dropout to estimate uncertainty necessitates multiple stochastic forward passes during adaptation, introducing notable computational overhead. Furthermore, the masking strategy employs a fixed masking ratio, without dynamically adjusting to evolving test-time conditions.

### 4.4 Conclusion

To complement the hierarchical taxonomy in Figure 3, Table 3 consolidates the key architectural and data assumptions of a few representative CTTA methods across the three introduced families. For each method, we report whether it maintains a memory buffer, employs a teacher-student framework, requires source statistics beyond the weights, relies on multiple augmentations, and is tied to a specific normalization layer type. These dimensions are particularly consequential in practice and real-time deployment.

## 5 Source Model Variants

The initialization of source models plays a crucial role and is the dominant paradigm in vision and other fields (Zoph et al., 2020). Since there is no access to the source data, the initial state of the model becomes the anchor in CTTA. While not a strict categorization, CTTA methods can be grouped based on the underlying model architecture. More recent approaches increasingly focus on continually adapting larger models such as Vision Transformers (ViT) (Dosovitskiy et al., 2021), which offer stronger representational capacity and improved generalization to unseen distribution shifts at test-time. That being said, a wave of CTTA methods (Wang et al., 2021; 2022; Park et al., 2024b; Maharana et al., 2025a; Schneider et al., 2020; Song et al., 2023; Hong et al., 2023; Wang et al., 2024a; Tian & Lyu, 2024; Chen et al., 2022; Niu et al., 2022; 2023) primarily employed CNN backbones as source models. For experiments on CIFAR-10C (Hendrycks & Dietterich, 2019), a pre-trained WideResNet28 (Zagoruyko & Komodakis, 2016) on CIFAR-10 is used as the source model. Similarly, on CIFAR-100C and ImageNet-C, pre-trained ResNeXt-29 (Xie et al., 2017) and ResNet-50 (He et al., 2016) are used for image classification tasks. These architectures provided a natural testbed for evaluating CTTA methods. Their reliance on BatchNorm layers also made them well-suited for update strategies like entropy minimization and the estimation of norm statistics, as discussed in previous sections. Another line of efforts have thus begun shifting toward transformer-based backbones like ViTs for CTTA (Liu et al., 2024b;a). Hybrid-TTA (Park et al., 2025) takes a different route and selects the best model fine-tuning strategy based on the test sample with a ViT backbone. In their paper, ViDA also runs experiments by continually adapting foundational models (Bommasani et al., 2021) at test-time. While full model fine-tuning is not advisable due to computational and overfitting concerns, ViDA enhances adaptation due to adapters being attached to the models. Typically, they perform experiments with DINOv2 (Oquab et al., 2024) and SAM (Kirillov et al., 2023) on CIFAR-10C and ImageNet-C to demonstrate the promise of this direction. Foundational models offer a key advantage in the CTTA setting: their large pretraining corpora enable strong generalization to unseen shifts.

## 6 The Dire Need for Online Continual Adaptation

The CTTA literature predominantly emphasizes *online* adaptation, wherein models must update continually as test data arrives in a streaming fashion. This paradigm is fundamentally distinct from offline adaptation,

Table 3: Representative CTTA methods and their key requirements and assumptions. **Mem.**: maintains a memory buffer or replay. **T-S**: uses a teacher-student framework. **Src.**: source statistics beyond the weights — **S**: BN statistics $(\mu_S, \sigma_S^2)$; **F**: Fisher information computed on source data; **R**: a small amount of source replay or covariance; −: source weights only. **Aug.**: requires multiple augmentations per test sample. **Norm.**: dependence on the backbone normalization layer — **BN**: requires BatchNorm; **LN**: requires LayerNorm; ⋆: norm-agnostic.

| | Method | Mem. | T-S | Src. | Aug. | Norm. |
|---|---|---|---|---|---|---|
| *Optimization-based* | *Entropy Minimization (§4.1.1)* | | | | | |
| | TENT (Wang et al., 2021) | – | – | – | – | BN |
| | EATA (Niu et al., 2022) | – | – | F | – | BN |
| | SAR (Niu et al., 2023) | – | – | – | – | ⋆ |
| | RMT (Döbler et al., 2023) | ✓ | ✓ | R | ✓ | ⋆ |
| | SoTTA (Gong et al., 2023) | ✓ | – | – | – | BN |
| | DeYO (Lee et al., 2024b) | – | – | – | ✓ | ⋆ |
| | *Pseudo-Labeling (§4.1.2)* | | | | | |
| | AdaContrast (Chen et al., 2022) | ✓ | – | – | ✓ | BN |
| | DSS (Wang et al., 2024a) | – | – | – | – | BN |
| | PLF (Tan et al., 2024) | – | – | – | – | BN |
| | RPL (Rusak et al., 2021) | – | – | – | – | BN |
| | *Topological Consistency (§4.1.3)* | | | | | |
| | TCA (Ni et al., 2025) | – | ✓ | – | – | – |
| | *Parameter Restoration (§4.1.4)* | | | | | |
| | PETAL (Brahma & Rai, 2023) | – | ✓ | F | ✓ | ⋆ |
| | RoTTA (Yuan et al., 2023) | ✓ | ✓ | S | – | BN |
| *Parameter-Efficient* | *Normalization Layers (§4.2.1)* | | | | | |
| | BN Stats Adapt (Schneider et al., 2020) | – | – | S | – | BN |
| | MixNorm (Hu et al., 2021) | – | – | S | ✓ | BN |
| | NOTE (Gong et al., 2022) | ✓ | – | – | – | BN |
| | MECTA (Hong et al., 2023) | – | – | – | – | BN* |
| | TTN (Lim et al., 2023) | – | – | S | – | BN |
| | *Adaptive Parameter Updates (§4.2.2)* | | | | | |
| | LAW (Park et al., 2024b) | – | – | – | – | ⋆ |
| | PALM (Maharana et al., 2025a) | – | – | – | – | ⋆ |
| | PSMT (Tian & Lyu, 2024) | – | ✓ | – | – | ⋆ |
| | FOA (Niu et al., 2024) | – | – | R | – | ⋆ |
| *Architecture-based* | *Teacher-Student (§4.3.1)* | | | | | |
| | CoTTA (Wang et al., 2022) | – | ✓ | – | ✓ | ⋆ |
| | C-CoTTA (Shi et al., 2025) | – | ✓ | ✓ | – | ⋆ |
| | CoDiRe (Chen et al., 2026b) | – | ✓ | – | – | ⋆ |
| | *Adapters (§4.3.2)* | | | | | |
| | EcoTTA (Song et al., 2023) | – | – | R | – | BN† |
| | ViDA (Liu et al., 2024b) | – | ✓ | R | ✓ | ⋆ |
| | Buffer (Kim et al., 2025) | – | – | – | – | ⋆ |
| | PAID (Wang et al., 2025c) | – | – | R | – | ⋆ |
| | GOLD (Lai et al., 2026) | – | ✓ | ✓ | – | ⋆ |
| | *Visual Prompting (§4.3.3)* | | | | | |
| | VDP (Gan et al., 2023) | – | ✓ | – | ✓ | ⋆ |
| | DPCore (Zhang et al., 2025d) | ✓ | – | R | – | ⋆ |
| | KFF (Zhou et al., 2025) | ✓ | – | R | – | ⋆ |
| | *Masked Modeling (§4.3.4)* | | | | | |
| | Continual-MAE (Liu et al., 2024a) | – | – | – | ✓‡ | ⋆ |

*MECTA replaces BN with MECTA-Norm. †EcoTTA's meta-networks contain BN layers but the source backbone can be arbitrary. ‡MC-Dropout uses multiple stochastic forward passes rather than data augmentations.

which assumes access to the entire target dataset for multiple training epochs (Wang et al., 2022; Zhu et al., 2024). In online CTTA, data is processed in a *single pass*; each batch is observed only once, predictions must be issued immediately, and the model cannot revisit past samples (Gong et al., 2022; Niu et al., 2022). This constraint reflects the operational reality of deployed systems where data arrives continuously, and storage of historical test samples may be infeasible due to memory limitations or privacy regulations.

**Practical Motivations.** Several compelling factors necessitate online adaptation in real-world deployments:

1. *Data Privacy*: In many applications, such as medical imaging or personal devices, retaining test data for iterative processing may violate privacy requirements. Online adaptation ensures that data is processed and discarded immediately (Mai et al., 2022).

2. *Real-Time Responsiveness*: Safety-critical applications like autonomous driving demand immediate predictions. An autonomous vehicle traveling at 60 mph covers approximately 30 meters during a one-second delay, making low-latency adaptation essential (Liu et al., 2019). Edge computing constraints further limit the feasibility of iterative optimization (Hong et al., 2023).

3. *Non-Stationary Environments*: Real-world distributions change continuously—weather conditions shift, lighting varies, and sensor characteristics drift. Multiple passes over stale data may actually harm performance as the underlying distribution evolves (Wang et al., 2022).

4. *Computational Constraints*: Edge devices and embedded systems have limited memory and compute budgets. Iterative adaptation requires storing gradients and intermediate states, which may exceed available resources (Song et al., 2023; Hong et al., 2023).

**Risks of Iterative Adaptation.** While iterative methods can achieve stronger adaptation under controlled conditions, they introduce significant risks in the CTTA setting. Repeated optimization on the same batch amplifies confirmation bias from noisy pseudo-labels, accelerating error accumulation (Wang et al., 2022). Furthermore, multiple gradient updates on test data lead to overfitting to transient domain characteristics, exacerbating catastrophic forgetting of source knowledge (Brahma & Rai, 2023). Empirical studies have shown that simply running TENT (Wang et al., 2021) with multiple update steps degrades long-term performance compared to single-step updates (Niu et al., 2022).

**Emerging Directions.** Recent work has begun exploring *on-demand* adaptation, which triggers model updates only when significant domain shifts are detected, rather than adapting on every batch (Ma et al., 2025). This paradigm offers a middle ground between continuous adaptation and static deployment, potentially reducing computational overhead while maintaining robustness. Additionally, forward-only adaptation methods that eliminate backpropagation entirely (Niu et al., 2024) represent a promising direction for resource-constrained deployment.

In summary, the practical focus of CTTA should remain on real-world deployment scenarios characterized by computational constraints, privacy requirements, and the need for immediate responsiveness to distributional shifts. The community has increasingly converged on single-pass, online adaptation as the standard evaluation protocol, reflecting these practical considerations.

# 7 Benchmarks & Experiments

In this section, we provide a comprehensive evaluation of CTTA methods across standard benchmarks. We discuss widely used datasets and evaluation protocols and present comparative results organized by task type: image classification and semantic segmentation. We report results directly from the original publications or from the unified benchmark studies (Wang et al., 2024b; Döbler et al., 2023) to ensure accuracy and reproducibility.

## 7.1 Datasets

### 7.1.1 Image Classification Benchmarks

**CIFAR-10-C / CIFAR-100-C.** Hendrycks & Dietterich (2019) introduced corruption benchmarks by applying 15 common image corruptions to the test sets of CIFAR-10 and CIFAR-100 (Krizhevsky et al., 2009). These corruptions simulate real-world perturbations and are applied at 5 severity levels. CIFAR-10-C and CIFAR-100-C each contain 10,000 corrupted test samples per corruption type, enabling standardized evaluation of model robustness under distribution shift.

**ImageNet-C.** Following the same corruption protocol, ImageNet-C (Hendrycks & Dietterich, 2019) applies 15 corruptions to the ImageNet (Deng et al., 2009) validation set, resulting in 50,000 images per corruption type across 1,000 classes. This larger-scale benchmark tests the scalability of CTTA methods to real-world model sizes and data complexity.

**ImageNet Variants.** Beyond synthetic corruptions, several datasets capture natural distribution shifts that complement the corruption benchmarks. ImageNet-R (Hendrycks et al., 2021a) contains 30,000 images across 200 ImageNet classes depicting rendition shifts including artistic representations, cartoons, and sketches. ImageNet-A (Hendrycks et al., 2021b) comprises natural adversarial examples that reliably cause classifier failures despite appearing benign to humans. ImageNet-Sketch provides sketch-based representations of ImageNet classes, testing robustness to texture removal.

**DomainNet.** DomainNet (Peng et al., 2019) contains ∼0.6 million images across 345 categories from 6 distinct visual domains (Clipart, Infograph, Painting, Quickdraw, Real, Sketch). DomainNet-126, a cleaned subset with 126 classes from 4 domains, is commonly used for CTTA evaluation under substantial domain gaps.

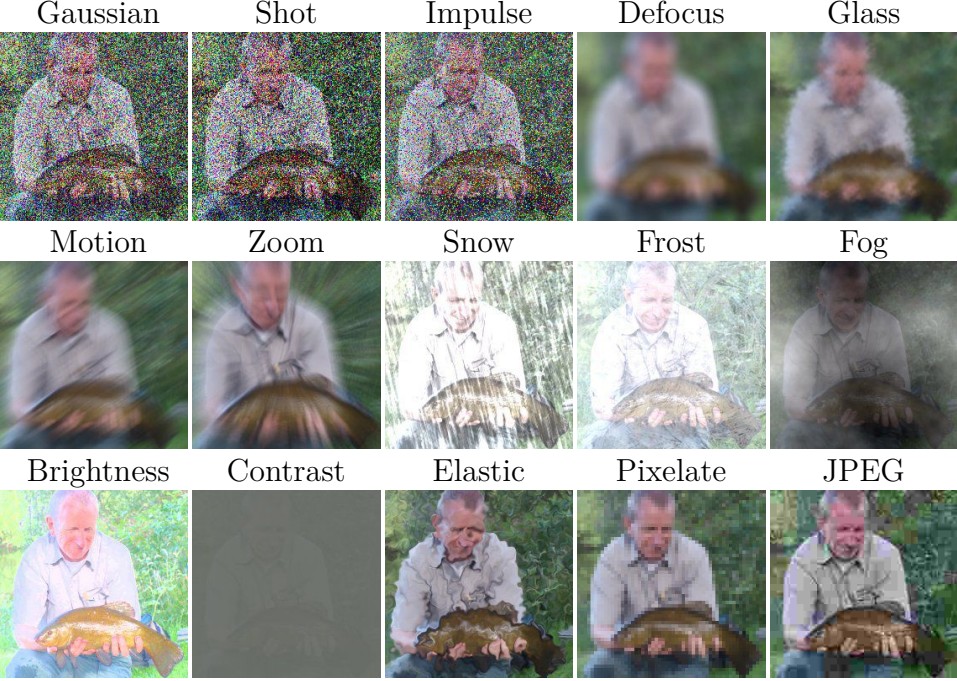

Figure 10: **ImageNet-C corruption examples.** The 15 corruption types are grouped into four categories: *Noise* (Gaussian, Shot, Impulse), *Blur* (Defocus, Glass, Motion, Zoom), *Weather* (Snow, Frost, Fog, Brightness), and *Digital* (Contrast, Elastic, Pixelate, JPEG).

### 7.1.2 Semantic Segmentation Benchmarks

**Cityscapes → ACDC.** The ACDC dataset (Sakaridis et al., 2021) provides semantic segmentation annotations for driving scenes under four adverse conditions: Fog, Night, Rain, and Snow. Models pre-trained on Cityscapes (Cordts et al., 2016) are adapted to these conditions in sequence, simulating real-world environmental changes encountered in autonomous driving.

**SHIFT Dataset.** SHIFT (Sun et al., 2022) is a synthetic driving dataset that simulates continuous domain shifts, including weather transitions and time-of-day changes, providing a challenging testbed for long-term CTTA evaluation.

## 7.2 Corruption Categories

The 15 corruptions in the standard benchmarks are organized into four categories based on their characteristics. *Noise* corruptions, comprising Gaussian, Shot, and Impulse noise, introduce random pixel-level perturbations that simulate sensor noise commonly encountered in low-light or high-ISO imaging conditions. *Blur* corruptions include Defocus, Glass, Motion, and Zoom blur, simulating various camera- and motion-induced distortions that arise from optical imperfections or relative movement between the camera and scene. *Weather* corruptions encompass Snow, Frost, Fog, and Brightness variations that mimic natural environmental conditions affecting outdoor imaging systems. Finally, *Digital* corruptions capture post-processing degradations through Contrast adjustment, Elastic Transform, Pixelation, and JPEG compression artifacts. These corruption types, designed to mirror real-world image degradations, provide a standardized framework for evaluating CTTA methods under diverse distribution shifts.

## 7.3 Evaluation Protocol

**CTTA Protocol.** Following (Wang et al., 2022), the standard CTTA evaluation protocol presents corruption types sequentially without a model reset between tasks. The model adapts online to each batch, and task boundaries are *unknown* to the model. This setup evaluates both adaptation capability and resistance to catastrophic forgetting.

**Metrics.** For classification tasks, we report the mean classification error (%) averaged across all corruptions at severity level 5 (the most challenging). For semantic segmentation, we report mean Intersection over Union (mIoU, %). Lower error rates and higher mIoU indicate better performance.

**Source Models.** To ensure fair comparison across methods, we report results using standardized source model architectures: WideResNet-28 (Zagoruyko & Komodakis, 2016) pre-trained on CIFAR-10 for CIFAR-10-C experiments, ResNeXt-29 (Xie et al., 2017) pre-trained on CIFAR-100 for CIFAR-100-C, ResNet-50 (He et al., 2016) pre-trained on ImageNet for ImageNet-C, and SegFormer-B5 (Xie et al., 2021) pre-trained on Cityscapes for the ACDC segmentation benchmark. Recent works additionally evaluate on Vision Transformer (ViT) (Dosovitskiy et al., 2021) backbones to assess method generalization beyond CNN architectures.

**Academic vs Realistic Evaluation Settings.** The standard CTTA evaluation protocol constitutes a *clean academic setting* that affords reproducibility and controlled comparison. Realistic deployment to continual shifts is different. First, the above benchmarks instantiate covariate shift (closed label space) exclusively under a fixed and finite set of 15 corruption types, whereas real-world shift is open-ended, often compound, and may involve shifts in label distribution, sensor characteristics, or scene semantics simultaneously. Second, the sequence order is deterministic and fixed. This means that methods implicitly exploit the predictable ordering. Third, severity level 5 represents the worst-case corruption intensity that is uniformly applied to the entire test stream. Fourth, the closed-set label assumption is structurally guaranteed by construction in corruption benchmarks, but not in real-time deployment. Readers should therefore interpret reported numbers as measures of performance under *controlled covariate shift with a fixed protocol* rather than as proxies for general-purpose continual adaptation capability.

Table 4: Mean classification error (%) on CIFAR-10-C under the CTTA protocol (severity level 5). All methods use WideResNet-28 unless noted. †ViT-Base backbone (Dosovitskiy et al., 2021). **Bold**: best in category; underline: best overall.

| Fam. | Method | Gaus. | Shot | Imp. | Def. | Glass | Mot. | Zoom | Snow | Frost | Fog | Brit. | Cont. | Elas. | Pix. | JPEG | Mean |
|---|---|---|---|---|---|---|---|---|---|---|---|---|---|---|---|---|---|
| | Source | 72.3 | 65.7 | 72.9 | 46.9 | 54.3 | 34.8 | 42.0 | 25.1 | 41.3 | 26.0 | 9.3 | 46.7 | 26.6 | 58.5 | 30.3 | 43.5 |
| EM | TENT (Wang et al., 2021) | 24.8 | 20.6 | 28.6 | 14.4 | 31.1 | 16.5 | 14.1 | 19.1 | 18.6 | 18.6 | 12.2 | 20.3 | 25.7 | 20.8 | 24.9 | 20.7 |
| | EATA (Niu et al., 2022) | 24.3 | 19.1 | 27.0 | 12.4 | 29.9 | 13.9 | 11.8 | 16.5 | 15.5 | 15.0 | 9.4 | 12.5 | 21.6 | 16.8 | 21.0 | 17.8 |
| | SAR (Niu et al., 2023) | 28.3 | 26.0 | 35.8 | 12.7 | 34.8 | 13.9 | 12.0 | 17.5 | 17.6 | 14.9 | 8.2 | 13.0 | 23.5 | 19.5 | 27.2 | 20.3 |
| | RMT (Döbler et al., 2023) | 21.7 | 18.6 | 24.2 | 10.3 | 24.0 | 11.2 | 9.5 | 12.1 | 11.7 | 10.3 | 7.0 | 8.7 | 14.8 | 10.5 | 14.5 | **13.9** |
| | SATA (Chakrabarty et al., 2024) | 23.9 | 20.1 | 28.0 | 11.6 | 27.4 | 12.6 | 10.2 | 14.1 | 13.2 | 12.2 | 7.4 | 10.3 | 19.1 | 13.3 | 18.5 | 16.1 |
| | SoTTA (Gong et al., 2023) | 23.1 | 19.2 | 26.8 | 11.8 | 26.5 | 12.8 | 10.5 | 14.2 | 13.5 | 12.0 | 7.6 | 10.1 | 18.2 | 12.8 | 17.8 | 15.8 |
| PL | DSS (Wang et al., 2024a) | 24.1 | 21.3 | 25.4 | 11.7 | 26.9 | 12.2 | 10.5 | 14.5 | 14.1 | 12.5 | 7.8 | 10.8 | 18.0 | 13.1 | 17.3 | 16.0 |
| | AdaContrast (Chen et al., 2022) | 29.1 | 22.5 | 30.0 | 14.0 | 32.7 | 14.1 | 12.0 | 16.6 | 14.9 | 14.4 | 8.1 | 10.0 | 21.9 | 17.7 | 20.0 | 18.5 |
| | PLF (Tan et al., 2024) | 23.5 | 18.7 | 23.6 | 10.4 | 24.4 | 10.9 | 10.6 | 12.7 | 11.9 | 10.4 | 8.0 | 9.7 | 16.4 | 12.0 | 16.2 | **14.8** |
| | RPL (Rusak et al., 2021) | 25.2 | 20.8 | 27.5 | 12.8 | 28.6 | 13.5 | 11.5 | 15.8 | 14.8 | 13.2 | 8.5 | 11.2 | 19.8 | 14.5 | 18.8 | 17.1 |
| PR | PETAL (Brahma & Rai, 2023) | 23.4 | 21.1 | 25.7 | 11.7 | 27.2 | 12.2 | 10.3 | 14.8 | 13.9 | 12.7 | 7.4 | 10.5 | 18.1 | 13.4 | 16.8 | 15.9 |
| | RoTTA (Yuan et al., 2023) | 22.5 | 19.8 | 24.2 | 10.8 | 25.1 | 11.5 | 9.8 | 13.2 | 12.5 | 11.2 | 7.2 | 9.5 | 16.5 | 11.8 | 15.2 | **14.7** |
| NL | BN Adapt (Schneider et al., 2020) | 28.1 | 26.1 | 36.3 | 12.8 | 35.3 | 14.2 | 12.1 | 17.3 | 17.4 | 15.3 | 8.4 | 12.6 | 23.8 | 19.7 | 27.3 | 20.4 |
| | NOTE (Gong et al., 2022) | 30.4 | 26.7 | 34.6 | 13.6 | 36.3 | 13.7 | 13.9 | 17.2 | 15.8 | 15.2 | 9.1 | 7.5 | 24.1 | 18.4 | 25.9 | 20.2 |
| | MECTA (Hong et al., 2023) | 26.5 | 22.8 | 30.2 | 12.2 | 31.5 | 13.2 | 11.8 | 15.8 | 15.2 | 13.8 | 8.2 | 11.5 | 21.2 | 16.5 | 22.8 | **18.2** |
| APU | LAW (Park et al., 2024b) | 24.7 | 18.9 | 25.5 | 12.9 | 26.7 | 15.0 | 11.8 | 15.1 | 14.7 | 15.9 | 10.1 | 13.8 | 19.4 | 14.7 | 18.3 | 17.2 |
| | PALM (Maharana et al., 2025a) | 25.8 | 18.1 | 22.7 | 12.3 | 25.3 | 13.1 | 10.7 | 13.5 | 13.1 | 12.2 | 8.5 | 11.8 | 17.9 | 12.0 | 15.4 | 15.5 |
| | PSMT (Tian & Lyu, 2024) | 22.8 | 18.9 | 23.2 | 11.2 | 24.4 | 12.3 | 10.2 | 13.7 | 13.0 | 11.4 | 7.8 | 9.5 | 16.2 | 11.8 | 15.4 | **14.8** |
| TTA-S | CoTTA (Wang et al., 2022) | 24.3 | 21.3 | 26.6 | 11.6 | 27.6 | 12.2 | 10.3 | 14.8 | 14.1 | 12.4 | 7.5 | 10.6 | 18.3 | 13.4 | 17.3 | **16.2** |
| VPAda. | ViDA† (Liu et al., 2024b) | 52.9 | 47.9 | 19.4 | 11.4 | 31.3 | 13.3 | 7.6 | 7.6 | 9.9 | 12.5 | 3.8 | 26.3 | 14.4 | 33.9 | 18.2 | 20.7 |
| | EcoTTA (Song et al., 2023) | 23.8 | 18.7 | 25.7 | 11.5 | 29.8 | 13.3 | 11.3 | 15.3 | 15.0 | 13.0 | 7.9 | 11.3 | 20.2 | 15.1 | 20.5 | **16.8** |
| | VDP (Gan et al., 2023) | 22.6 | 19.7 | 28.1 | 7.1 | 28.4 | 9.5 | 6.3 | 10.2 | 11.5 | 9.0 | 1.5 | 5.6 | 18.5 | 12.8 | 18.5 | **14.0** |
| MMI | C-MAE† (Liu et al., 2024a) | 30.6 | 18.9 | 11.5 | 10.4 | 22.5 | 13.9 | 9.8 | 6.6 | 6.5 | 8.8 | 4.0 | 8.5 | 12.7 | 9.2 | 14.4 | **12.6** |

## 7.4 Image Classification Results

### 7.4.1 CIFAR-10-C Results

Table 4 presents comprehensive results on CIFAR-10-C. Methods are organized by their primary strategy as discussed in § 4.

*Key Observations.* The source model's 43.5% mean error highlights the severity of the distribution shift introduced by corruptions. Among optimization-based methods, RMT achieves the best CNN-based result (13.9%) by combining robust mean teacher updates with entropy minimization. Parameter restoration approaches also prove effective: RoTTA attains 14.7% error through robust batch normalization and category-balanced sampling that handles temporally correlated test streams. Architecture-based methods show particular promise, with Continual-MAE achieving the overall best performance (12.6%) by leveraging masked image modeling on a ViT backbone. Notably, VDP demonstrates that visual prompting can achieve competitive results (14.0%) with minimal parameter overhead, offering an efficient alternative to full model adaptation.

### 7.4.2 CIFAR-100-C Results

Table 5 presents results on the more challenging CIFAR-100-C benchmark with 100 classes.

*Key Observations.* The increased number of classes (100 vs. 10) makes pseudo-label quality more critical, as evidenced by the larger performance gaps between methods. RoTTA and VDP achieve the best results in the 28-29% error range, demonstrating the importance of robust normalization and parameter-efficient adaptation strategies. Pure normalization-based methods such as BN Stats Adapt and NOTE show larger degradation compared to their CIFAR-10-C performance, revealing their limitations when scaling to more complex classification tasks with finer-grained distinctions between classes.

### 7.4.3 ImageNet-C Results

Table 6 presents results on the large-scale ImageNet-C benchmark.

*Key Observations.* The source model error of 82.0% is significantly higher than on CIFAR benchmarks, reflecting the increased difficulty of large-scale classification under corruption. RMT achieves the best reported result (54.8%), demonstrating the scalability of teacher-student approaches with robust mean updates to ImageNet-scale models. Memory-efficient methods, including MECTA and EcoTTA, achieve competitive

Table 5: Mean classification error (%) on CIFAR-100-C (severity level 5) using ResNeXt-29. EM=Entropy Min., PL=Pseudo-Label, PR=Param. Restoration, NL=Norm. Layers, APU=Adaptive Param., T-S=Teacher-Student, VP=Visual Prompt, Ada.=Adapters.

| Method | Cat. | Error (%) |
|---|---|---|
| Source | – | 46.5 |
| TENT (Wang et al., 2021) | EM | 34.2 |
| EATA (Niu et al., 2022) | EM | 31.8 |
| SAR (Niu et al., 2023) | EM | 32.5 |
| RMT (Döbler et al., 2023) | EM | **29.4** |
| SoTTA (Gong et al., 2023) | EM | 30.2 |
| DSS (Wang et al., 2024a) | PL | 30.8 |
| AdaContrast (Chen et al., 2022) | PL | 32.1 |
| PLF (Tan et al., 2024) | PL | **29.8** |
| CoTTA (Wang et al., 2022) | T-S | 30.5 |
| PETAL (Brahma & Rai, 2023) | PR | 30.2 |
| RoTTA (Yuan et al., 2023) | PR | **28.9** |
| BN Adapt (Schneider et al., 2020) | NL | 35.8 |
| NOTE (Gong et al., 2022) | NL | **33.5** |
| LAW (Park et al., 2024b) | APU | 31.2 |
| PALM (Maharana et al., 2025a) | APU | 29.5 |
| PSMT (Tian & Lyu, 2024) | APU | **29.2** |
| VDP (Gan et al., 2023) | VP | **28.5** |
| EcoTTA (Song et al., 2023) | Ada. | 30.8 |

Table 6: Mean classification error (%) on ImageNet-C (severity level 5) using ResNet-50 backbone. EM=Entropy Min., NL=Norm. Layers, APU=Adaptive Param., T-S=Teacher-Student, Ada.=Adapters.

| Method | Cat. | Error (%) |
|---|---|---|
| Source | – | 82.0 |
| TENT (Wang et al., 2021) | EM | 62.5 |
| EATA (Niu et al., 2022) | EM | 58.8 |
| SAR (Niu et al., 2023) | EM | 57.2 |
| RMT (Döbler et al., 2023) | EM | **54.8** |
| CoTTA (Wang et al., 2022) | T-S | 56.2 |
| BN Adapt (Schneider et al., 2020) | NL | 65.2 |
| NOTE (Gong et al., 2022) | NL | 60.5 |
| MECTA (Hong et al., 2023) | NL | **58.2** |
| LAW (Park et al., 2024b) | APU | 56.8 |
| PALM (Maharana et al., 2025a) | APU | **55.2** |
| EcoTTA (Song et al., 2023) | Ada. | 57.5 |

performance while significantly reducing computational overhead, validating their suitability for practical deployment scenarios where resources are constrained.

## 7.5 Semantic Segmentation Results

Table 7 presents results on the Cityscapes → ACDC benchmark for semantic segmentation.

*Key Observations.* All methods exhibit the largest performance degradation on Night conditions, indicating the particular difficulty of adapting to extreme illumination changes that fundamentally alter image statistics.

Table 7: Semantic segmentation (mIoU %) on Cityscapes → ACDC over 10 rounds with a SegFormer-B5 backbone (Xie et al., 2021).

| Method | Fog | Night | Rain | Snow | Mean |
|---|---|---|---|---|---|
| Source | 69.2 | 40.3 | 59.8 | 57.5 | 56.7 |
| BN Adapt (Schneider et al., 2020) | 68.5 | 38.2 | 58.5 | 55.8 | 55.3 |
| TENT (Wang et al., 2021) | 67.8 | 37.5 | 57.2 | 54.2 | 54.2 |
| CoTTA (Wang et al., 2022) | 71.5 | 42.8 | 62.5 | 60.2 | 59.3 |
| RMT (Döbler et al., 2023) | 72.2 | 43.5 | 63.8 | 61.5 | 60.3 |
| SVDP (Park et al., 2024a) | 72.8 | 44.2 | 64.5 | 62.0 | 60.9 |
| DAT (Ni et al., 2024) | 73.5 | 44.8 | 65.2 | 62.8 | 61.6 |
| Hybrid-TTA (Park et al., 2025) | 74.2 | 45.5 | 66.0 | 63.2 | **62.2** |

Critically, TENT and BN Stats Adapt perform *worse* than the unadapted source model on this benchmark because SegFormer employs LayerNorm rather than BatchNorm, rendering BN-centric adaptation strategies ineffective. Teacher-student methods, including CoTTA, RMT, and their variants, achieve consistent improvements by leveraging pseudo-label refinement and temporal consistency constraints. Among recent methods, Hybrid-TTA achieves state-of-the-art performance (62.2% mIoU) while operating approximately 20× faster than comparable methods (Park et al., 2025), demonstrating that computational efficiency and accuracy need not be mutually exclusive in segmentation CTTA.

## 7.6 Analysis and Discussion

### 7.6.1 Performance vs. Computational Cost

The experimental results reveal a fundamental trade-off between adaptation accuracy and computational requirements that practitioners must carefully consider when deploying CTTA methods. Teacher-student frameworks such as CoTTA and RMT consistently achieve strong performance across benchmarks, but this comes at the cost of approximately doubled memory consumption due to maintaining two network copies. For applications where computational resources are abundant, these methods represent the most reliable choice.

At the other end of the spectrum, methods specifically designed for efficiency offer compelling alternatives. EcoTTA reduces memory overhead through lightweight meta-networks while maintaining competitive accuracy. MECTA introduces memory-economic normalization that enables adaptation on edge devices with limited GPU memory. FOA eliminates backpropagation through forward-only adaptation, dramatically reducing computational requirements at the cost of modest accuracy reduction. Between these extremes, methods like EATA and SAR achieve a balanced profile by selectively filtering samples before adaptation, reducing unnecessary gradient computations while preserving accuracy on informative samples.

### 7.6.2 Long-Term Adaptation Stability

A critical yet often underexplored aspect of CTTA evaluation is performance stability over extended adaptation periods. Our analysis of multi-round experiments reveals stark differences between methods. TENT, when run continuously without model reset, exhibits progressive error accumulation that can exceed 50% classification error after 10 rounds of sequential corruptions on CIFAR-10-C. This degradation stems from the compounding effect of noisy pseudo-labels and the absence of any mechanism to preserve source knowledge.

In contrast, methods with explicit forgetting prevention demonstrate remarkable stability. CoTTA and PETAL maintain consistent performance through stochastic parameter restoration, periodically reverting a subset of weights to their source values. RoTTA achieves similar stability through a different mechanism: category-balanced sampling ensures that the memory bank maintains a representative distribution of classes, preventing the model from drifting toward majority classes in temporally correlated streams. These findings underscore that long-term deployment scenarios demand careful consideration of forgetting mitigation strategies beyond single-round evaluation metrics.

### 7.6.3 Sensitivity to Hyperparameters

CTTA methods exhibit varying degrees of sensitivity to hyperparameter choices, which have important implications for practical deployment where extensive tuning is infeasible. Learning rate selection proves critical across all methods: values in the range of $10^{-3}$ to $10^{-4}$ typically yield stable adaptation, while higher rates accelerate initial adaptation but risk catastrophic overfitting to early batches. Batch size requirements differ substantially across method families. Normalization-based approaches generally require batches of 64 or larger to obtain reliable statistical estimates, whereas instance-normalization methods like NOTE can operate effectively with batch sizes as small as 1, making them suitable for streaming scenarios. For teacher-student frameworks, the EMA momentum coefficient $\lambda$ governs the trade-off between teacher stability and responsiveness to distribution changes; most methods adopt $\lambda = 0.999$ as a robust default that prevents teacher collapse while allowing gradual adaptation.

### 7.6.4 Which Conclusions are Robust vs. Benchmark Dependent?

**Conclusions that are robust.** The failure of BatchNorm-centric methods on transformer architectures (where BatchNorm is absent) is a structural result that holds regardless of the benchmark. TENT and BN Adapt underperform the source model in Table 7. The observation that pseudo-label quality becomes more critical as the number of classes grows, as evidenced in performance gaps in Tables 4 and 5, reflects a fundamental property of self-training under class imbalance rather than a benchmark-specific effect.

**Conclusions that are benchmark dependent.** For instance, the strong performance of teacher-student frameworks such as RMT in Table 4 is partly attributable to the large, i.i.d. batches of size 200 used in the standard protocol, which are favorable for mean teacher stability. Throughout, all classification results are obtained under the CSC protocol with deterministic corruption ordering; performance under CDC (Zhang et al., 2025d) is reported only by a subset of methods and is generally worse, meaning that methods evaluated only under CSC should be treated as having an incomplete robustness profile.

### 7.7 Summary and Practical Recommendations

The comprehensive evaluation across classification and segmentation benchmarks yields several actionable insights for practitioners deploying CTTA systems. First, method selection should be guided by the underlying model architecture: BatchNorm-centric methods excel on CNN backbones but fail on transformer architectures that employ LayerNorm, where adapter-based and prompting methods generalize more reliably. Second, teacher-student frameworks consistently achieve the strongest results across diverse benchmarks, making them the default recommendation when memory constraints permit. Third, for resource-constrained deployment on edge devices, EcoTTA, MECTA, and FOA provide favorable accuracy-efficiency trade-offs that enable real-time adaptation. Fourth, applications requiring long-term stability must incorporate explicit forgetting prevention mechanisms, whether through parameter restoration, category-balanced sampling, or teacher anchoring. Finally, semantic segmentation under extreme domain shifts, such as nighttime conditions remains substantially more challenging than classification, with even state-of-the-art methods showing significant performance gaps that indicate opportunities for future research.

## 8 Emerging Trends and Future Directions

The rapid evolution of CTTA research has opened several promising avenues for future investigation. In this section, we identify key emerging trends and outline directions that we believe will shape the next generation of continual adaptation methods.

### 8.1 Beyond Vision: CTTA for Other Modalities and Downstream Tasks

While the majority of CTTA research has focused on image classification and segmentation (Wang et al., 2022), real-world systems increasingly operate across multiple modalities, domains, and downstream tasks. Beyond vision, CTTA has begun attracting attention in natural language processing (NLP) tasks. Liu et al. (2025b) studies CTTA for text understanding. Distributional shifts in text alter both the surface

form and the semantic content. The absence of standardized NLP benchmarks analogous to ImageNet-C remains a significant barrier to progress, and developing CTTA methods that are architecture-agnostic is an important open direction. CTTA has also been gaining traction for medical imaging problems (Valanarasu et al., 2024; Zhao et al., 2026; Du et al., 2026; Ji et al., 2026) where scanner hardware variation, acquisition protocol differences, or patient population heterogeneity could be causes of shifts. To add on, CTTA for 3D tasks involving human pose estimation (Peng et al., 2026), semantic segmentation (Cao et al., 2023), and point cloud understanding (Jiang et al., 2024a) has been explored. Autonomous systems, robotics, and augmented reality applications rely on heterogeneous sensor inputs, including cameras, LiDAR, radar, and audio (Wang et al., 2025d; Lin et al., 2024). Distribution shifts in these settings often affect modalities differently: fog may degrade camera inputs while leaving LiDAR largely unaffected, whereas rain introduces noise across both. Current CTTA methods, designed primarily for single-modality image data, struggle to exploit cross-modal complementarity or handle modality-specific shifts. There has been rising CTTA works for audio-visual data (Wang et al., 2025b; Zhang et al., 2025b; Maharana et al., 2026) involving single and bimodal distributional shifts. We believe future research should develop fusion-aware adaptation strategies that dynamically reweight modality contributions based on estimated reliability and mechanisms for graceful degradation when individual modalities become unreliable. CTTA has also been studied and integrated for federated learning (Wang et al., 2026) and cloud-edge (Xu et al., 2026).

The emergence of vision-language models (VLMs) such as CLIP (Radford et al., 2021) and foundation models like GPT-4 presents new opportunities and challenges for CTTA (Wang et al., 2025a). Recent works, including TDA (Karmanov et al., 2024) have begun exploring training-free adaptation of VLMs through dynamic caching mechanisms. However, adapting models with billions of parameters at test-time remains computationally prohibitive. Promising directions include prompt-based adaptation that modifies only the input space, adapter modules that add minimal trainable parameters, and retrieval-augmented approaches that leverage external knowledge bases without model updates. On similar lines, Jangamreddy et al. (2026) study the adaptation of visual foundational models like DinoV2 (Oquab et al., 2024) for segmentation in adverse weather conditions.

Beyond the Euclidean structure of data that is commonly studied, TTA has also emerged for graphs (Chen et al., 2026a; Qiao et al., 2025). The inherent challenges include the introduction of distribution shifts at test-time, across time, which affect node information, context, and thereby the structure of the graph. In addition, from a model deployment perspective, it is critical to adapt a small subset of parameters of a graph neural network (GNN). There has been recent work on continual test-time training for graphs (Cai et al., 2026), but limited extensions towards CTTA for graphs. This represents a significant gap, as graph-structured data is ubiquitous in real-world applications, including social networks, molecular property prediction, traffic forecasting, and knowledge graphs. Hence, there is a need to develop more principled CTTA methods for graphs by carefully handling shifts and preserving the graph structure.

Across all of these settings, a common bottleneck is the absence of standardized benchmarks. Progress in non-vision CTTA will likely require the community to invest in domain-specific evaluation protocols before algorithmic advances can be reliably measured and compared.

## 8.2 Continual Adaptation for LLMs and Multimodal LLMs

An emerging and largely unexplored direction is extending CTTA to large language models (LLMs) and multimodal LLMs (MLLMs). A key parallel exists between CTTA in vision and recent efforts in test-time reinforcement learning for LLMs. In the vision setting, CTTA methods must adapt without ground-truth labels, relying instead on self-generated signals such as entropy or pseudo-labels. Similarly, recent work on test-time RL for LLMs faces the challenge of obtaining reward signals without verified answers (Liu et al., 2025a; Zhao et al., 2025b; Zhang et al., 2025a; Fu et al., 2025). TTRL (Zuo et al., 2025) proposes using majority voting over multiple sampled outputs as a proxy reward to train LLMs via RL on unlabeled test data, achieving strong gains on mathematical reasoning benchmarks. Related efforts explore self-rewarding mechanisms where the model itself provides feedback, as well as self-play and process-based reward estimation to reduce reliance on human annotations. These approaches share a core motivation with CTTA: enabling models to improve from unlabeled, potentially shifted test data at deployment time.

However, a key limitation of existing test-time RL methods is that they operate under a *single-domain* assumption. TTRL and related approaches evaluate on a fixed task distribution (e.g., one math benchmark) and do not consider the scenario where the task domain evolves. In practice, both LLMs and MLLMs encounter continually changing inputs: a deployed reasoning model may face mathematical problems, then code generation tasks, then scientific questions, each with different distributional characteristics (Zhao et al., 2025a; Che et al., 2026). Similarly, a multimodal model may encounter shifting visual domains alongside varied query types. This is directly analogous to the CTTA setting, where the test distribution is non-stationary, and task boundaries are unknown. Applying CTTA principles to LLM and MLLM adaptation, such as preventing catastrophic forgetting of prior capabilities while adapting to the current domain, represents a promising direction. The intersection of self-supervised reward estimation and continual, domain-shifting deployment remains open and could benefit from the insights developed in the CTTA literature surveyed in this paper.

### 8.3 Black-Box Adaptation in the Real-world

Most existing CTTA methods assume full white-box access to model internals, including parameters, gradients, and intermediate activations. However, this assumption is increasingly misaligned with how models are deployed in practice. Recent backpropagation-free methods such as FOA (Niu et al., 2024) and TDA (Karmanov et al., 2024) improve efficiency by eliminating gradient computation, but they still require access to internal tokens, features, or normalization statistics. These approaches are better described as *gray-box*: they reduce computational cost while relying on partial model access.

A fundamentally different setting arises when the model is entirely opaque, accessible only through an input-output API. In this *strict black-box* setting, users can only submit a raw input and receive the output prediction probabilities; the model's architecture, parameters, pre-training data, and all intermediate representations remain unknown. This is increasingly common as powerful models are deployed behind commercial APIs, where each query also incurs monetary cost and network latency. Unlike the gray-box case, black-box adaptation cannot leverage any internal signal and must rely solely on the mapping between inputs and output distributions. While adapting black-box models has received attention in the supervised transfer learning literature (Tsai et al., 2020; Oh et al., 2023; Zhang et al., 2026a;b), where labeled support sets are available to guide prompt learning or zeroth-order optimization, the unsupervised setting relevant to TTA and CTTA remains largely open. Existing approaches either refine output predictions post-hoc but offer limited adaptive capacity, or modify the input through augmentation, purification, or zeroth-order prompt learning, each facing notable trade-offs between query cost, latency, and stability under unsupervised signals (Boudiaf et al., 2022; Gao et al., 2023; Nie et al., 2022). No existing method achieves both strong adaptation performance and practical efficiency in this strict setting. Developing query-efficient and stable black-box adaptation strategies, particularly under the continual distribution shifts of CTTA, is an important and timely direction.

### 8.4 Adaptation for Foundation Models

The scale of modern foundation models, with parameters numbering in the billions, fundamentally changes the adaptation landscape. Both open-source models (LLaMA (Touvron et al., 2023), Mistral) and proprietary systems (GPT-4 (Achiam et al., 2023), Claude) present distinct challenges. Full fine-tuning of billion-parameter models is impractical for test-time scenarios due to memory, compute, and latency constraints. Parameter-efficient techniques, including LoRA (Hu et al., 2022), adapters, and prompt tuning, offer paths to lightweight adaptation by modifying only a small fraction of parameters. Extending these approaches to the continual setting, where efficiency must be maintained across sequential domain shifts, remains an open challenge. Key questions include how to prevent adapter drift over time, whether different domains require separate adapter modules, and how to compose adaptations from multiple domains. Not all layers or components contribute equally to domain-specific performance. Building on insights from surgical fine-tuning (Lee et al., 2023b) and layer-wise importance estimation (Park et al., 2024b; Maharana et al., 2025a), future methods should dynamically identify and adapt only the most relevant model components for each shift. Hierarchical approaches that first adapt early layers for low-level feature shifts and later layers for semantic shifts could provide both efficiency and effectiveness.

Large language models exhibit remarkable in-context learning capabilities (Dong et al., 2024), adapting behavior based on prompts or demonstrations without any parameter updates. Extending this paradigm to continual settings, where context must be managed across evolving distributions, offers a fundamentally different approach to adaptation. Challenges include maintaining context relevance as distributions shift, preventing context contamination from outlier samples, and balancing context length against computational cost.

### 8.5 Robustness Under Adversarial and Pathological Shifts

Current CTTA benchmarks primarily evaluate natural corruptions and domain shifts. However, deployed systems may encounter adversarial perturbations, out-of-distribution samples, or pathological edge cases that exploit adaptation mechanisms. Adaptation mechanisms that update model parameters based on test inputs create potential attack surfaces. An adversary could craft inputs designed to corrupt the adapted model, causing failures on subsequent benign samples. Understanding the vulnerability of CTTA methods to such attacks and developing robust adaptation strategies that resist adversarial manipulation represents a critical direction for safety-critical applications. Most CTTA methods assume that test samples, while shifted, belong to known classes. In practice, test streams may contain novel classes, anomalies, or samples far outside the training distribution. Integrating out-of-distribution detection with adaptation, such that the model can identify when not to adapt, would prevent corruption from irrelevant or harmful samples. Methods should balance the plasticity needed for adaptation with the stability required to reject inappropriate updates.

### 8.6 Standardized Benchmarks for Realistic Evaluation

The predominant reliance on synthetic corruption benchmarks (CIFAR-C, ImageNet-C) limits our understanding of CTTA performance under realistic conditions. Future benchmark development should address several gaps. Real-world shifts exhibit complex temporal patterns including gradual drift, abrupt transitions, cyclic variations, and combinations thereof. Benchmarks should systematically evaluate methods across this spectrum of dynamics. The SHIFT dataset (Sun et al., 2022) represents progress in this direction, but broader coverage of shift types and application domains is needed. Practical deployment often involves strict constraints on memory, compute, latency, and energy consumption. Benchmarks should incorporate resource budgets as first-class evaluation criteria, measuring not only accuracy but adaptation efficiency. This would encourage the development of methods suitable for edge devices, mobile platforms, and real-time applications.

Most current evaluations span tens of domain shifts at most. Understanding method behavior over hundreds or thousands of shifts, as would occur in long-running deployed systems, requires new evaluation protocols. Lifelong learning benchmarks that test knowledge retention, forward transfer, and stability over extended periods would better reflect deployment realities.

### 8.7 Theoretical Foundations

Despite empirical progress, the theoretical understanding of CTTA remains limited. Several fundamental questions merit investigation. The tension between adapting to new distributions and retaining source knowledge is central to CTTA, yet formal characterization of this trade-off is lacking. Theoretical analysis connecting adaptation rate, forgetting rate, and shift magnitude would provide principled guidance for method design and hyperparameter selection. How many samples are required to reliably adapt to a new distribution? Under what conditions do CTTA methods converge to good solutions versus collapse? Establishing sample complexity bounds and convergence guarantees would strengthen the foundations of the field and identify fundamental limits of adaptation. CTTA intersects with online learning, continual learning, domain adaptation, and robust optimization. Formalizing these connections could enable the transfer of theoretical tools and insights across communities, accelerating progress on shared challenges.

# 9 Conclusion

Distribution shifts are an inevitable consequence of deploying machine learning systems in dynamic, real-world environments. CTTA addresses this challenge by enabling models to adapt online to shifting distributions using only unlabeled test data, without access to source training data or knowledge of when shifts occur.

In this survey, we have provided a comprehensive review of the CTTA landscape. We began by establishing the problem formulation and distinguishing CTTA from related paradigms, including standard test-time adaptation, domain adaptation, and continual learning. We then presented a systematic taxonomy organizing methods into three primary families: optimization-based approaches that design self-training objectives through entropy minimization, pseudo-labeling, and parameter restoration; parameter-efficient methods that adapt normalization statistics or selectively update model subsets; and architecture-based approaches that introduce teacher-student frameworks, adapters, visual prompts, or masked modeling objectives.

Our experimental analysis across standard benchmarks revealed several key insights. Teacher-student frameworks consistently achieve strong performance but at the cost of doubled memory requirements. Parameter restoration and category-balanced sampling prove essential for long-term stability. Normalization-based methods, while efficient, fail on architectures without BatchNorm layers. Semantic segmentation under extreme conditions like nighttime remains substantially more challenging than classification. These findings provide practitioners with guidance for selecting appropriate methods based on their deployment constraints and requirements.

Looking ahead, we identified emerging trends including adaptation for multi-modal systems and vision-language models, frameworks that operate across white-box and black-box settings, parameter-efficient approaches for billion-scale foundation models, and robustness under adversarial conditions. We also highlighted the need for standardized benchmarks that better reflect real-world deployment scenarios with diverse shift dynamics, resource constraints, and long-horizon evaluation.

The field of CTTA has matured rapidly since CoTTA (Wang et al., 2022) introduced the paradigm in 2022, with dozens of methods now addressing various aspects of the challenge. Yet significant open problems remain, particularly regarding theoretical foundations, scalability to massive models, and deployment in safety-critical applications. We hope this survey serves as both a comprehensive reference for current approaches and a roadmap for future research toward building machine learning systems that gracefully adapt to the ever-changing conditions of the real world.

**Acknowledgement**

We would like to thank the anonymous reviewers for their helpful comments. This project was partially funded by The University of Texas at Dallas Office of Research and Innovation through the SPIRe grant program. This research was also partially supported by the National Science Foundation (NSF) under Grant No. 2513070.

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

# A    Broader Impact Statement

Our comprehensive survey on continual test-time adaptation (CTTA) focuses on self-adapting models to continual target or test distributions. While this capability addresses genuine and pressing challenges in robust deployment, it also introduces a set of ethical, safety, and societal considerations that we discuss below.

**Safety-Critical Deployment.**  A recurring motivation is the applicability of CTTA to safety-critical domains that include autonomous driving, medical imaging, and robotic perception. In these domains, the consequences of erroneous predictions can directly affect human safety and well-being. CTTA methods adapt models at test-time using unsupervised signals, which are inherently noisy and can degrade under severe or unexpected distribution shifts. This central challenge of error accumulation means that such a system can silently drift toward degraded performance over time, potentially without any observable external indication. We urge practitioners deploying CTTA in safety-critical settings to maintain rigorous human oversight, establish out-of-distribution detection mechanisms to identify when adaptation should be suspended, and define clear performance thresholds beyond which the system falls back to a validated static model or requests human intervention.

**Medical Imaging.**  The deployment of continually self-adapting models in medical imaging warrants severe caution. Distribution shifts may arise from changes in scanner hardware, imaging protocols, or patient population demographics. However, the absence of ground-truths can reinforce errors as the model might adapt toward confident but incorrect predictions. Any clinical deployment of CTTA systems must be subject to regulatory oversight, prospective clinical validation, and mechanisms ensuring that adaptation does not compromise diagnostic reliability. The results reported in this survey are drawn from standard computer vision benchmarks and should not be taken as evidence of clinical readiness.

**Privacy Implications.**  CTTA methods adapt models using real deployment data. Even though test data retention is not done due to data privacy regulations, gradient-based adaptation can cause the model to memorize statistical properties of test samples in its parameters, creating a potential privacy risk if the adapted model is later inspected. A good future research direction is to understand the memorization properties of any CTTA method before deployment in privacy-sensitive contexts, and consider adaptation strategies that provably limit the influence of individual samples on updated parameters.

**Benchmark Limitations and Generalization Claims.**  The benchmarks widely adopted by the CTTA research community may not faithfully represent the shift dynamics, severity levels, or data characteristics of real-world deployment scenarios. We encourage the community to invest in domain-specific benchmarks with clinically and operationally validated shift scenarios before advocating for deployment in high-stakes settings.

**Positive Societal Impacts.**  Notwithstanding the concerns above, this survey addresses a genuine and important problem. Machine learning systems deployed in the real world inevitably encounter data that differs from their training distribution, and a model that can adapt gracefully to such shifts is more reliable, longer-lived, and less dependent on costly periodic retraining. Advances in CTTA have direct positive potential for improving the robustness of diagnostic tools in resource-limited clinical settings, enabling safer autonomous systems that handle diverse environmental conditions, and reducing the computational and environmental cost of model redeployment. Realizing this potential responsibly requires that the research community treat the safety, accountability, and robustness dimensions of CTTA as integral parts of the research agenda rather than post-deployment considerations.

