# OpenReview forum: "Continual Test-Time Adaptation in Computer Vision: Methods, Benchmarks, and Future Directions"
_TMLR — Accepted by TMLR_

### Review · Reviewer_Cm9J · 2026-03-26

**Summary Of Contributions:**

This paper presents a broad survey of continual test-time adaptation (CTTA). It formalizes the setting, clarifies the main challenges such as catastrophic forgetting and error accumulation, and organizes existing methods into a clear taxonomy spanning optimization-based, parameter-efficient, and architecture-based approaches. The paper also reviews common evaluation protocols and benchmarks, and closes with a discussion of open problems and future directions.

**Additional Comments:**

Overall, I found this to be a useful and well-organized survey on an important and active topic. The paper would be stronger with a slightly sharper critical perspective and a brief acknowledgment of nearby developments outside the core vision setting, but I think it already provides value in its current form.

**Audience:**

Yes

**Audience Explanation:**

For a survey paper, the main claims are largely about organization, coverage, and synthesis of the field, and these are mostly supported. The paper clearly defines CTTA, motivates the problem well, and presents a structured taxonomy with representative methods and benchmark settings. I also found the discussion of domain-shift patterns and open problems helpful. My main reservation is not about correctness, but about completeness at the boundaries of the stated scope.

**Broader Impact Concerns:**

No major additional concerns beyond the standard risks already associated with deployment-time adaptation. As with related work in this area, there is some risk that adaptation under distribution shift can fail silently or reinforce errors over time, so a brief note on reliability in safety-critical settings would be sufficient.

**Claims And Evidence:**

Yes

**Claims Explanation:**

Strengths
1. CTTA has grown quickly, and a focused survey is valuable.
2. The taxonomy and problem setup make the literature easier to navigate.
3. Objectives, adaptation patterns, representative methods, and evaluation settings are all discussed coherently. The paper is generally readable and should be a useful entry point for researchers new to this area.

Weaknesses
1. The scope is still fairly vision-centric, so the discussion of adjacent domains is limited.
2. Some parts are more descriptive than critical; a stronger synthesis of what has and has not worked would make the survey more insightful.

**Requested Changes:**

1. The discussion of benchmarks and evaluation protocols could be sharper. In particular, it would help to distinguish clean academic settings from more realistic continual shifts and to clarify which conclusions appear robust versus benchmark-dependent.

2. A brief discussion of CTTA-related developments beyond vision would make the survey more complete, like the scenarios in graph learning. For example, the authors should consider mentioning:

Chen et al., Test-Time Adaptation for Graph Learning: A Systematic Survey (2026)

Qiao et al., GCAL: Adapting Graph Models to Evolving Domain Shifts (ICML 2025)

3. Some sections would benefit from a bit more synthesis beyond method listing. Short end-of-section takeaways on what seems effective, what remains fragile, and what the main trade-offs are would improve the paper.

4. A compact summary table of representative methods and their assumptions would improve usability, e.g., whether they rely on memory buffers, teacher models, source statistics, multiple augmentations, or specific normalization layers.

---

> ### Author Response · Authors · 2026-06-02
> **Response to Reviewer Cm9J**
>
> We are glad that Reviewer Cm9J found our survey valuable, well-motivated, and easy to navigate. Below, we respond to the critical points raised in the review.
>
> 1. > The discussion of benchmarks and evaluation protocols could be sharper.
>
> **Response:**
> We thank the reviewer for this important suggestion. In response, we have made two additions to Section 7, marked in the revised manuscript in orange.
>
> In the first addition, we have added a paragraph titled *Academic vs. Realistic Evaluation Settings* to Section 7.3. We draw an explicit and principled distinction between what the standard CTTA evaluation protocol actually measures and what real-world continual deployment demands. The discussion is along 4 dimensions, i.e., a fixed and finite set of corruption types, the deterministic corruption ordering, the uniform corruption severity level 5 assumption, and the guaranteed closed-set assumption. We conclude the paragraph by explicitly cautioning readers that all experimental results are under controlled settings and assumptions. A general-purpose protocol would be different.
>
> In the second addition, we have introduced a new subsection 7.6.4 titled *Which Conclusions are Robust vs. Benchmark Dependent?*. This subsection separates the empirical findings into two categories. Conclusions that appear robust across benchmarks include the failure of BatchNorm-centric methods in transformer architectures and the decline of pseudo-label quality as the number of classes grows. This reflects in the results as well. Conclusions identified as benchmark-dependent include the performance rankings of teacher-student frameworks under large i.i.d. batches. We also discuss why CSC protocols should be treated as incomplete.
>
>
> 2. > Discussions beyond vision and addition of graph-learning scenarios.
>
> **Response:**
> Thank you for pointing this out. We would like to mention that we have narrowed down the survey's scope by changing the title to *Continual Test-Time Adaptation in Computer Vision: Methods, Benchmarks, and Future Directions*. We have also extended Section 8.1 (in blue). The section title has been revised to *Beyond Vision: CTTA Across Modalities and Downstream Tasks*. In this expanded discussion, we examine the unique challenges associated with applying CTTA to domains such as NLP, medical imaging, and 3D perception tasks, including human pose estimation, semantic segmentation, and point-cloud segmentation. We additionally highlight emerging directions in multimodal CTTA, particularly for audio-visual learning.
>
> As suggested, we also incorporate our discussions on TTA for graph learning. We discuss the data and model learning complexities of TTA on graphs and note that CTTA methods for graphs are limited.
>
>
> 3. > Some sections would benefit with end-of-section takeaways.
>
> **Response:**
> We thank you for this suggestion. In response to this, to each major subsection in Section 4, we have discussions focusing on the effectiveness, fragility, and trade-offs of each. By this, we go beyond pure method listing and discussion. To Section 4.1 for optimization-based methods, we have added a new consolidated paragraph called *Discussions* (in orange). On similar lines, in our previous draft, to each subsection (4.2.1, 4.2.2, 4.3.1, 4.3.2, 4.3.3, 4.3.4) in Sections 4.2 and 4.3, we have had similar discussion paragraphs. We hope these additions help readers develop a complete intuition and principled considerations.
>
>
>
> 4. > A compact summary table of representative methods and their assumptions would improve usability.
>
> **Response:**
> This is a very good suggestion. We summarize all the methods based on reliance on buffers, teacher models, source statistics, multiple augmentations, or specific normalization layers. To our revised draft, we have added a new subsection (Section 4.4, orange text) and table (Table 3, orange text), and compared all the representative CTTA methods that are mentioned in our work. We hope this improves usability and presents a unified view.
>
> 5. > Broader impact concern.
>
> **Response:**
> We thank the reviewer for raising this.  In response, we have added a dedicated Broader Impact Statement to the manuscript, where we address model deployment in safety-critical settings. Please find it after the references.
>
>
> *We appreciate your thoughtful comments and suggestions. We hope our responses address your concerns. Please let us know if there are any additional questions, and we will be happy to discuss further.*

---

> > ### Comment · Reviewer_Cm9J · 2026-06-06
> > **Response to the revision**
> >
> > Thank you for your detailed revision. This paper has been well improved, so I recommend acceptance.

---

> > > ### Author Response · Authors · 2026-06-06
> > >
> > > Thank you for giving a thumbs up! We really appreciate it!

---

### Review · Reviewer_Q6JW · 2026-03-29

**Summary Of Contributions:**

This manuscript presents a comprehensive survey of Continual Test-Time Adaptation (CTTA), a paradigm in which a pre-trained model adapts on-the-fly to a non-stationary stream of unlabeled target distributions without access to source data or task boundaries. The authors formally define the CTTA problem, analyze diverse continual domain shift patterns , and propose a hierarchical taxonomy that organizes existing methods into three families: (1) optimization-based strategies (entropy minimization, pseudo-labeling, topological consistency, parameter restoration), (2) parameter-efficient methods (normalization layer adaptation, adaptive parameter selection), and (3) architecture-based approaches (teacher-student frameworks, adapters, visual prompting, masked modeling). The survey reviews representative methods within each category, presents comparative benchmarks on CIFAR-10-C, CIFAR-100-C, ImageNet-C, and Cityscapes$\to$ACDC, and discusses emerging directions including foundation model adaptation, multi-modal CTTA, LLM/MLLM adaptation, and black-box settings.

**Audience:**

Yes

**Audience Explanation:**

Yes.The topic is squarely within TMLR's scope, and the audience interest is broad and well-motivated for several reasons including rapidly growing subfield with no dedicated survey. Timely coverage of emerging directions.  Section 8 discusses CTTA for foundation models.

**Broader Impact Concerns:**

The manuscript does not include a Broader Impact Statement. While a formal statement is not strictly required for a  survey paper, this particular survey actively promotes the deployment of self-adapting models in safety-critical domains (autonomous driving, medical imaging, robotic perception) without adequately discussing the ethical and safety implications of doing so.

**Claims And Evidence:**

Yes

**Claims Explanation:**

The taxonomy and the benchmarking tables are well-supported and internally consistent. The experimental results appear to be faithfully aggregated from the original papers. However:

- The characterization of VDP as a teacher-student method (Section 4.3.1, p. 17) is  factually inaccurate  and contradicted by the authors' own description of VDP in Section 4.3.3 (p. 20).
- The distinction between CTTA and standard Continual Learning, while discussed, is not drawn with sufficient precision to be convincing to a reader unfamiliar with both fields.
-  The distributional shift taxonomy (Table 1) makes formal claims but provides no concrete evidence or examples to ground them, significantly weakening the survey's pedagogical value. Section 2.3 introduces four types of distributional shift --- covariate, concept, conditional, and label --- and Table 1 encodes each as a set of probabilistic equalities and inequalities (e.g., covariate shift: $p(x_s) \neq p(x_t)$, $p(y_s|x_s) = p(y_t|x_t)$). While mathematically precise, these conditions are presented in a vacuum: there is not a single concrete scenario, visual illustration, or dataset-grounded example anywhere in the section that would help a reader *recognize* these shifts in practice.

**Requested Changes:**

**S1. Add concrete examples to Table 1.**
Table 1 currently lists only formal probabilistic conditions for each shift type. Add a column with intuitive, real-world visual examples:
- **Covariate shift:** Model trained on clear-weather driving images deployed in fog/snow (same classes, altered visual appearance).
- **Concept shift:** Relationship between features and labels changes --- e.g., "road surface" features that predicted "safe" in dry training conditions now appear wet at test time.
- **Conditional shift:** Input distribution conditioned on labels changes --- e.g., "car" images predominantly sedans in training but trucks at test time.
- **Label shift:** Class distribution changes --- e.g., balanced healthy/diseased in training, deployed in high-prevalence screening region.

**S2. Define or explicitly exclude "semantic shift."**
The term "semantic shift" is well-known in the OOD detection and open-world recognition literature (referring to encountering classes unseen during training) but is absent from the taxonomy in Section 2.3. Either (i) define it and explain its relationship to the four listed shift types, or (ii) explicitly state why it is excluded from scope.


**S3. Clarify BatchNorm update mechanics (Section 4.2.1, p. 13).**
Clearly distinguish between the two types of BatchNorm parameters and their update mechanisms:
- **Running statistics** ($\mu$, $\sigma^2$): updated via forward passes using exponential moving average. Not learned parameters; do not receive gradients. At test time, methods like TENT replace these with batch-level statistics.
- **Affine parameters** ($\gamma$, $\beta$): the **only** learnable parameters updated via backpropagation on the entropy/self-training loss. Constitute <1% of total model parameters.
The current text presents Equations 10--11 for affine updates but does not clearly state that running statistics are updated through a fundamentally different mechanism. This conflation may mislead readers.

**S4. Explain the ViDA adapter rank asymmetry rationale (Section 4.3.2, p. 19).**
Explicitly state the architectural design rationale:
- **High-rank adapters** $\to$ greater representational capacity $\to$ assigned to capture stable, domain-agnostic knowledge shared across distributions. High rank encodes rich, persistent representations.
- **Low-rank adapters** $\to$ limited capacity (implicit regularization) $\to$ assigned to capture transient, domain-specific shifts. Enables fast adaptation without overfitting to any single domain.

The uncertainty-based reweighting via MC Dropout dynamically shifts emphasis: high uncertainty upweights the stable high-rank pathway; low uncertainty lets the adaptive low-rank pathway contribute more. This rationale is central to ViDA's design.

**S5 Inconsistency in VPD method**
VDP (Gan et al., 2023) is described in Section 4.3.1 as a teacher-student weight-update method, when it is in fact a visual prompting method that keeps the backbone entirely frozen. The authors themselves correctly describe VDP in Section 4.3.3 and illustrate it in Figure 8, creating an internal contradiction.

**S6 Citation Issues**
these are the issues that I have discovered, there might be some others:
- Mishra et al. --- cited as **2026** | Paper is arXiv:2411.17002, posted **November 2024**. The year 2026 is fabricated/projected. Must be corrected to 2024. (Shambhavi Mishra, Julio Silva-Rodriguez, Ismail Ben Ayed, Marco Pedersoli, and Jose Dolz. Semantic
anchor transport: Robust test-time adaptation for vision-language models, 2026. URL https://arxiv.
org/abs/2411.17002. page 36)
-  Lim et al. --- cited as **ICCV 2023** | TTN was published at **ICLR 2023**, not ICCV 2023. Wrong conference venue.  (Hyesu Lim et al. Ttn: A domain-shift aware batch normalization in test-time adaptation. In ICCV, 2023. in page 35)

---

> ### Author Response · Authors · 2026-06-02
> **Response to Reviewer Q6JW**
>
> We really appreciate Reviewer Q6JW for giving us very detailed feedback on our survey. Thank you for finding our work well-motivated and broad. We address all of the concerns and requested changes below. We have also revised our draft, and all requested changes are marked in red.
>
> 1. > Distinction between standard continual learning and CTTA.
>
> **Response:**
> This is a great point-out and suggestion. For someone new to both fields, this will help in clearly understanding the differences between the two learning paradigms. To our revised draft in Section 2, we have added a new subsection (Section 2.2) by the name of *Continual learning vs. CTTA*, which systematically compares the two along six dimensions. In this, we compare based on *Supervision*, *The role of source model*, *Data access and replay*, *Compute regime*, *Task boundary information*, and *Objective asymmetry*. To provide a quick summary/reference, we also added a table consolidating our response (see Table 1).
>
> 2. > Add concrete examples to Table 1.
>
> **Response:**
> Thank you for this suggestion. We agree that such a column would help the reader to visualize each distributional shift. To make Table 1 comprehensive (now Table 2), we have added a new column *Example*. To each distribution shift, we have added the suggested detailed examples that motivate CTTA throughout the paper.
>
> 3. > Define or explicitly exclude "semantic shift"
>
> **Response:**
> We thank the reviewer for raising this point. Semantic shift is an important concept in the OOD detection and open-world recognition literature. In the revised manuscript, we have added a dedicated paragraph (now Section 2.4) - *On Semantic Shift and Scope*. We first define semantic shift precisely. We then explicitly state that semantic shift is outside the scope of this survey, as virtually all CTTA methods surveyed herein presuppose a closed label space. We do, however, acknowledge the growing body of work on open-set TTA and on settings involving joint covariate and label shifts, and note that extending CTTA to the open-set regime is largely an open research direction. We hope this addition clarifies the intended scope.
>
> 4. > Clarify BatchNorm update mechanics (Section 4.2.1, p. 13).
>
> **Response:**
> Thank you for pointing out this clear observation. In the revised draft, we have restructured Section 4.2.1 to explicitly distinguish between the two types of BatchNorm parameters and their update mechanisms. Running statistics are now introduced as non-trainable quantities maintained via an exponential moving average. We have added two new equations (now Eqns. 9,10) to reflect the same. These are then replaced by batch-level estimates $\mu_t$ and $\sigma^2_t$ through forward passes alone, with no gradient computation involved. Affine parameters ($\gamma$, $\beta$), that are trainable, are updated via the entropy loss. We also update Eqn. 9 (now Eqn. 11) to denote the transformed input feature.
>
> 5. > Explain the ViDA adapter rank asymmetry rationale.
>
> **Response:**
> Thank you for the suggestion. The suggested comment will strengthen the discussion on ViDA and its design choice on adding low-rank and high-rank adapters. We consolidated the suggested points and have added the required sentences to Section 4.3.2.
>
> 6. > Factual inaccuracy and inconsistency of VDP.
>
> **Response:**
> We sincerely apologize for the confusion and contradiction that has been caused. As mentioned, we plan on removing our highlighted discussion on VDP in Section 4.3.1 as a teacher-student model and instead would completely describe it as a visual-prompting method (Section 4.3.3 & Figure 8). This way, the sole focus would be on VDP as a visual-prompting CTTA algorithm.
>
>
> 7. > Citation issues.
>
> **Response**
> Thank you for bringing this to our notice. We sincerely apologize for this. We have made all the required citation changes throughout the revised draft.
>
>
> 8. > Addition of a Broader Impact Statement.
>
> **Response:**
> We thank the reviewer for raising this concern.  In response, we have added a dedicated Broader Impact Statement to the manuscript. Please find it after the references. First, we discuss the risks specific to safety-critical deployment, especially in autonomous driving and medical imaging contexts. Second, we address medical imaging specifically and how standard CTTA benchmark results should not be taken as proof of clinical readiness. Third, we talk about privacy implications. We focus on how parameter updates over sensitive test data could lead to memorization and need further investigation. Finally, we acknowledge the positive societal impacts of CTTA.
>
>
> *We appreciate your thoughtful comments and suggestions. We hope our responses address your concerns. Please let us know if there are any additional questions, and we will be happy to discuss further.*

---

### Review · Reviewer_sz1k · 2026-05-23

**Summary Of Contributions:**

This paper is a survey on Continual Test-Time Adaptation (CTTA), where a pretrained model adapts online to continuously changing unlabeled targets. The challenges are catastrophic forgetting, where continual updates erase source knowledge, and error accumulation, where unreliable signals degrade the model. The paper categorizes existing CTTA methods into three main families: optimization-based, parameter-efficient, and architecture-based approaches. They show that CTTA performance depends not only on the adaptation objective but also on the domain shift pattern, source architecture, batch composition, temporal correlation, and resource constraints. Future CTTA research should therefore move beyond simple corruption-based benchmarks toward more realistic settings involving dynamic shifts, recurring domains, foundation models, black-box systems, and safety-critical deployment.

**Audience:**

Yes

**Audience Explanation:**

The paper is relevant to researchers working on test-time adaptation, robustness, continual learning, and deployment reliability, as it organizes existing methods, evaluation settings, and open challenges in one survey. Its discussion of shift patterns and practical limitations would also be useful for practitioners considering adaptive models in dynamic environments.

**Broader Impact Concerns:**

No concerns.

**Claims And Evidence:**

Yes

**Claims Explanation:**

Strengths
* The paper addresses CTTA, an increasingly important problem for deploying models under non-stationary distribution shifts without source data or target labels.
* The survey organizes CTTA methods into three families: optimization-based, parameter-efficient, and architecture-based approaches, which help readers understand the field systematically.
* The paper explains different continual shift patterns such as CSC, gradual shifts, PTTA, CDC, and recurring domains, highlighting that CTTA performance depends heavily on the evaluation protocol.

Weaknesses
* The taxonomy and discussion look like a catalog of existing methods rather than a deeply critical synthesis with a new conceptual framework.
* The title suggests a comprehensive CTTA survey, but the paper mainly focuses on vision-based recognition and only briefly discusses other areas.
* It will be more compelling if it more explicitly compares methods by their assumptions, failure modes, computational costs, source-model requirements, robustness under realistic deployment conditions, etc.

**Requested Changes:**

* The paper claims to be a comprehensive CTTA survey, but the focus is mostly on vision-based benchmarks. The authors should either narrow the claim or expand the discussion of other non-vision domains such as NLP, medical imaging, 3D perception, vision-language models, etc.
* Many sections read like a catalog of existing methods. The authors should more explicitly compare methods in terms of their assumptions, failure modes, computational cost, memory requirements, and robustness under different shift patterns.
* The paper notes that most CTTA benchmarks focus on covariate shifts caused by image corruptions. The authors should clearly state and justify only considering the covariate shift, or discuss other shifts, such as conditional shift, label shift, semantic shift, recurring shifts, or adversarial shifts.

---

> ### Author Response · Authors · 2026-06-02
> **Response to Reviewer sz1k**
>
> We sincerely thank Reviewer sz1k for their constructive feedback and for finding our survey relevant, useful, and addressing an important problem. We address the concerns below.
>
> 1. > The paper claims to be a comprehensive CTTA survey, but the focus is mostly on vision-based benchmarks.
>
> **Response:**
> Thank you for this very important close observation. We want to point out that most papers in the CTTA research community have been on vision benchmarks involving CNN and ViT source model architectures. However, as suggested, narrowing the scope of this work would be a better option. We have updated the paper title to *Continual Test-Time Adaptation in Computer Vision: Methods, Benchmarks, and Future Directions*. We believe that the title now encapsulates all the important discussions.
> In addition, we have expanded our discussions on non-vision domains (marked in blue) in Section 8.1. We have updated the section title to *Beyond Vision: CTTA for other Modalities and Downstream Tasks*. In this, we characterize the distinct CTTA challenges posed by NLP, medical imaging, and 3D tasks like human pose estimation, semantic segmentation, and point-cloud segmentation. We also discuss rising multimodal CTTA for vision-language and audio-visual. We also touch upon existing works on graph learning, as well. We conclude with issues that must be addressed for CTTA to mature beyond its current vision-centric foundations.
>
> 2. > Many sections read like a catalog of existing methods.
>
> **Response:**
> We thank the reviewer for this important critique. We would like to mention that our rationale for dividing CTTA methods is based on what each method adapts and its adaptation objective. In each subsection of Section 4, we critically analyze and compare each CTTA method. At the end of each, we also have a discussion/limitation section to summarize our takeaways with respect to effectiveness, failures, and trade-offs. However, in response to the suggestion, we have also made a few targeted changes in the revised draft.
>
> Assumptions - Table 3 consolidates the key assumptions of all representative methods. We cover memory buffer, teacher-student dependencies, source data requirement, multiple augmentations per test sample, and norm layer dependency.
>
> Computational cost and memory requirements - Section 7.6.1 provides a dedicated analysis of the performance vs computational cost trade-off. Table 3 also has memory requirements implicitly through the memory buffer and teacher-student columns.
>
> Robustness under different shift patterns - While most CTTA methods specifically address covariate shift (please see the next answer for more), in Section 7.6.4, we explicitly separate findings that hold across evaluation protocols and those that are dependent on benchmarks specifically.
>
>
> 3. > The authors should clearly state and justify only considering the covariate shift, or discuss other shifts.
>
> **Response:**
> We thank the reviewer for raising this point and concern. Yes, a large number of works in CTTA focus on addressing covariate shifts. We note that a similar remark has been made in the context of the standard TTA literature [1] as well. We believe that stems from benchmark availability and the algorithmic assumptions underlying self-training/unsupervised objectives used at test-time.
>
> In response to additional shifts, please see Section 2.4, where we probabilistically discuss and compare covariate, conditional, concept, and label shifts. To add on, in our draft of Table 2, we have added an additional column, *Example*, to give real-world examples. This should aid the reader in connecting the formulae to real-life examples. To broaden our discussion, we have added a new paragraph on *Semantic Shift and Scope* (in red). Since semantic shift is often observed in OOD and open-world learning scenarios, we discussed this in detail and compare against other shifts. We have also explicitly mentioned the scope of this survey to be limited to covariate shifts due to a closed-set assumption. We have also mentioned a few recent works on addressing open-set TTA. However, CTTA for addressing such shifts remains an open problem.
>
> In Section 3, as in the previous draft, we have critically discussed recurring and evolving domain shifts. We also discuss different domain dynamics and patterns that one might encounter at test-time.
>
> [1] Zehao Xiao and Cees GM Snoek. Beyond model adaptation at test time: A survey. arXiv preprint arXiv:2411.03687, 2024.
>
>
> *We appreciate your thoughtful comments and suggestions. We hope our responses address your concerns. Please let us know if there are any additional questions, and we will be happy to discuss further.*

---

### Decision · Action_Editor_kTUy · 2026-06-28

**Recommendation:** Accept with minor revision

**Additional Comments:**

As mentioned above, this paper provides a comprehensive and accurate survey of the rapidly growing field of CTTA with a focus on advances in the computer vision domain. Its clear presentation and taxonomy of different problem settings, approaches, advantages, and limitations, together with its broad coverage of the relevant literature, should benefit a broad TMLR audience. Two reviewers and I recommend awarding this paper the Survey Certification.

**Requests:**
If this paper is accepted, I request the authors to prepare the camera ready version based on the latest revision to incorporate the discussions with the reviewers. Also, please use correct punctuations around equations. Please clearly define all math symbols/letters, including $P_T$, without expecting the reader to guess what they mean. Currently some of them are unclear. For example, the meanings of $p$ in Eqs. (4, 5, etc.) and $\mathcal{H}$ in Eqs. (5, 6, 12, 13, etc.) are unclear, especially because these letters seem to be used in multiple places for different roles. The variable being optimized is unclear in argmax of Eq. (6). Spaces are missing around some of the $\neq$ symbols. Some of double quotes symbols look inconsistent and unusual (did you use `` and ''?). Does Eq. (22) correctly close the parens? Please consider using \mathrm{}, \text{}, or \textrm{}, or \textup{} for words used in equations such as $\mu_{\textup{mixed}}$. I encourage the authors take time to double-check each equation through the paper.

**Audience:**

Yes

**Audience Explanation:**

The reviewers all agree that Continual Test-Time Adaptation (CTTA) is an important and rapidly emerging field of machine learning. This paper provides a comprehensive survey of the field, along with insightful discussions of its limitations and open challenges. It should serve as a valuable resource for researchers seeking an overview of the existing literature as well as for those interested in contributing to the field.

**Claims And Evidence:**

Yes

**Claims Explanation:**

The reviewers positively evaluated the overall accuracy of the survey, its thorough grounding in the original references, its comprehensive taxonomy, and its coherent discussions from multiple perspectives. A few concerns were raised by the reviewers:
- Organization of the paper (Reviewer sz1k & Cm9J-R3)
- Adjustment of claims, in particular on the scope (Reviewer sz1k & Cm9J)
- Requested additions for enhancement (Reviewer Q6JW-S1 & S2, Reviewer Cm9J-R2 & R4)
- Clarifications (Reviewer Q6JW-S3 & S4, Reviewer Cm9J-R1)
- Inaccurate descriptions and citations (Reviewer Q6JW-S5 & S6)

The authors successfully addressed these concerns by revising the manuscript: improving the writing, reorganizing several sections, and incorporating the suggested enhancements.

---

> ### Author Response · Authors · 2026-07-09
>
> We sincerely thank the AE and all reviewers for their valuable time, thoughtful feedback, and constructive suggestions. We greatly appreciate the detailed comments, which have helped improve the quality and clarity of our manuscript. We hope our responses have addressed the reviewers’ concerns satisfactorily. As noted by the AE, the majority of the requested revisions had already been incorporated in the revised manuscript. The final camera-ready version (now uploaded) includes all remaining action items requested by the AE.
>
> Changes in the Camera-Ready Version
>
> 1. Improved Writing and Presentation
>
> We carefully revised the manuscript to improve its overall presentation. Specifically, we:
>
> * Corrected punctuation and formatting throughout the paper.
> * Redefined symbols and equations wherever clarification was needed.
> * Removed redundant text and improved readability.
> * Addressed all remaining formatting inconsistencies.
>
> 2. Updated Related Work
>
> To further strengthen the paper, we expanded the manuscript by discussing several recent papers published in CVPR 2026, ICML 2026, and TMLR 2026.
>
> CVPR 2026
>
> * The Golden Subspace: Where Efficiency Meets Generalization in Continual Test-Time Adaptation
> * Dance Across Shifts: Forward-Facilitation Continual Test-Time Adaptation through Dynamic Style Bridging
> * Back to Source: Open-Set Continual Test-Time Adaptation via Domain Compensation
> * Towards Stable Federated Continual Test-Time Adaptation in the Wild World
> * Cross-Architecture Adaptation: Cloud-Edge Continual Test-Time Adaptation with Dynamic Sampling and Heterogeneous Distillation
>
> CVPR 2026 Findings
>
> * Test-Time Distillation for Continual Model Adaptation
> * Wake the Sleeping Weights: Sparsely-Activated Continual Test-Time Adaptation for Medical Image Segmentation
> * Continual Adaptation of Vision Foundation Models for Semantic Segmentation in Adverse Weather
>
> ICML 2026
>
> * Blocking the Leakage: Manifold-Aware Gradient Projection for Long-Horizon Test-Time Adaptation
>
> TMLR 2026
>
> * Family Matters: A Systematic Study of Spatial vs. Frequency Masking for Continual Test-Time Adaptation